# Incomplete Multimodality-Diffused Emotion Recognition

**Yuanzhi Wang, Yong Li, Zhen Cui**
PCA Lab, Key Lab of Intelligent Perception and Systems for High-Dimensional
Information of Ministry of Education, School of Computer Science and Engineering,
Nanjing University of Science and Technology, Nanjing, China.
{yuanzhiwang, yong.li, zhen.cui}@njust.edu.cn

## Abstract

Human multimodal emotion recognition (MER) aims to perceive and understand human emotions via various heterogeneous modalities, such as language, vision, and acoustic. Compared with unimodality, the complementary information in the multimodalities facilitates robust emotion understanding. Nevertheless, in real-world scenarios, the missing modalities hinder multimodal understanding and result in degraded MER performance. In this paper, we propose an Incomplete Multimodality-Diffused emotion recognition (IMDer) method to mitigate the challenge of MER under incomplete multimodalities. To recover the missing modalities, IMDer exploits the score-based diffusion model that maps the input Gaussian noise into the desired distribution space of the missing modalities and recovers missing data abided by their original distributions. Specially, to reduce semantic ambiguity between the missing and the recovered modalities, the available modalities are embedded as the condition to guide and refine the diffusion-based recovering process. In contrast to previous work, the diffusion-based modality recovery mechanism in IMDer allows to simultaneously reach both distribution consistency and semantic disambiguation. Feature visualization of the recovered modalities illustrates the consistent modality-specific distribution and semantic alignment. Besides, quantitative experimental results verify that IMDer obtains state-of-the-art MER accuracy under various missing modality patterns. Codes are released at https://github.com/mdswyz/IMDer.

## 1 Introduction

Benefiting from the intrinsic heterogeneity of multimodal data, various modalities are exploited for multimodal emotion recognition (MER) to understand human behaviors and intents from a collaborative perspective [27, 13, 36]. Recently, MER has become one of the most active research topics of affective computing with many applications, such as healthcare [3, 18] and robotics [11, 12].

Robust MER relies on learning and combining representations from diverse modalities [27]. In previous studies, Zadeh *et al.* [31] designed a Tensor Fusion Network that takes as input paired modalities to encode the bimodal representation, which are further fused to generate the trimodal representation. Tsai *et al.* [27] proposed a multimodal transformer to learn the potential adaptation and correlations between modalities. Subsequently, various advanced approaches [17, 15, 13] have explored different variants of multimodal transformers to construct robust MER frameworks.

However, in real-world scenarios, not all modalities are consistently available, e.g., language data may be missing due to speech recognition errors; video data may be inaccessible due to privacy

---

Corresponding authors: Yong Li, Zhen Cui

37th Conference on Neural Information Processing Systems (NeurIPS 2023).

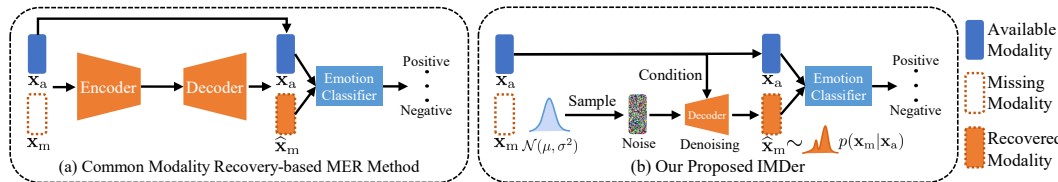

Figure 1: (a) shows the convenient modality recovery-based MER method. Typically, a well-crafted encoder-decoder architecture is used to recover the missing modalities. (b) illustrates our proposed Incomplete Multimodality-Diffused emotion recognition (IMDer) method. IMDer exploits the vanilla available modalities an the condition to guide and refine the distribution space of the recovered modality, thus achieves distribution consistency and semantic disambiguation.

and security concerns [14]. Ultimately, these incomplete multimodal data significantly hinder the performance of MER. For MER under incomplete multimodalities, one straightforward way is to recover the missing modalities from the available ones. As shown in Fig. 1 (a), convenient modality recovery methods [20, 35, 14, 30] aim to recover the missing modalities via a well-designed encoder-decoder framework and build a mapping between the vanilla available and the missing modalities. Among them, Zhao *et al.* [35] combined autoencoder with cycle consistency learning for modality recovery. Lian *et al.* [14] designed a graph completion network that utilized graph neural networks to reconstruct missing parts. However, these previous methods fail to explicitly consider the modality-specific distributions that are highly correlated with each modality's intrinsic discriminability, e.g., an image gives the visual appearance of a happy face via thousands of pixels, whilst the corresponding text describes this emotion with a sentence using discrete words.

In this paper, we aim to challenge MER under incomplete multimodalities by proposing an Incomplete Multimodality-Diffused emotion recognition (IMDer) method, as shown in Fig. 1 (b). To recover the missing modalities, IMDer exploits the prevalent score-based diffusion model [24] that maps the input random noise into the distribution space of the missing modalities. In particular, the score-based diffusion model captures the distribution of the missing modalities by perturbing data with the stochastic differential equation (SDE). With sufficient data and model capacity, we are capable of exploiting the well-trained score model to recover the distribution-consistent modality by solving reverse-time SDE (i.e., denoising process) starting from the prior noise distribution.

To reduce the semantic ambiguity between the missing and the corresponding recovered modalities, we use the vanilla available modalities as semantic conditions to guide and refine the recovering process. The mere information embedded in the available modalities facilitates IMDer simultaneously reaching both distribution consistency and semantic disambiguation. Finally, the recovered modalities together with available modalities could be jointly fed into a multimodal fusion and prediction network for MER. In summary, the contributions of this work can be concluded as:

- To address the challenge of MER under incomplete multimodalities, we propose the Incomplete Multimodality-Diffused emotion recognition (IMDer) method. IMDer maps input random noise to the distribution space of missing modalities and recovers missing data in accordance with their original distributions.

- To minimize the semantic ambiguity between the missing and recovered modalities, we utilize the available modalities as prior conditions to guide and refine the recovering process. This ensures that the recovered modality is both distribution and semantic-consistent.

- We perform extensive experiments on publicly available MER datasets and achieve superior or comparable results across different missing modality patterns. The feature visualization of the recovered modalities demonstrates consistent distribution and semantic alignment.

## 2  Background

### 2.1  Incomplete Multimodal Learning

Incomplete multimodal learning is an essential research topic in multimodal machine learning, as it can deal with the inevitable modality-missing environments in real-world scenarios. One effective approach is to find a low-dimensional subspace shared by all modalities, in which different modalities

have maximum correlation with each other [6, 2, 29]. However, this approach may ignore the complementarity between heterogeneous modalities, leading to suboptimal results for downstream tasks. Another more effective approach is to explicitly recover the missing modalities based on available ones. Representative methods include zero-based recovery [19], average-based recovery [34], and deep learning-based recovery [20]. Of these, the zero and the average-based recovery methods still cause a considerable gap between the recovered and the original data because they do not utilize any supervision information. In contrast, deep learning-based methods can better estimate missing modalities by leveraging their powerful feature representation capabilities. For example, Tran *et al.* [26] recovered missing modalities with a cascaded residual autoencoder. Pham *et al.* [20] and Zhao *et al.* [35] aimed to learn the joint multimodal representation via the cross-modality recovery strategy with cycle consistency loss for missing modalities issue. Lian *et al.* [14] utilized graph neural networks [4] to reconstruct missing modalities to improve MER accuracy.

## 2.2 Score-based Generative Diffusion Model

The score-based generative model is a family of generative models [22–24], which models data distribution through the Stein score function [25, 16]. Given a probability density function $p(\mathbf{x})$, the Stein score function (abbreviated as score) is defined as $s(\mathbf{x}) = \nabla_{\mathbf{x}} \log p(\mathbf{x})$. As we might guess, the score-based generative model aims to parameterize a score network $s(\mathbf{x}; \theta)$ with learnable parameters $\theta$ and train it to estimate $\nabla_{\mathbf{x}} \log p(\mathbf{x})$. Unlike likelihood-based models such as normalizing flows [10], score-based models do not have to be normalized and are easier to parameterize.

**Perturbing data with a diffusion process.** To generate samples with score-based models, we need to consider a diffusion process that corrupts data slowly into random noise. Let $\{\mathbf{x}(t) \in \mathbb{R}^d\}_{t=0}^T$ be a diffusion process, indexed by the continuous time variable $t \in [0, T]$. From the landmark work [24], a diffusion process is built by a stochastic differential equation (SDE), formally,

$$d\mathbf{x} = \mathbf{f}(\mathbf{x}, t)dt + g(t)d\mathbf{w}, \tag{1}$$

where $\mathbf{f}(\cdot, t)$ denotes the drift coefficient of the SDE, $g(t)$ is called the diffusion coefficient, and $\mathbf{w}$ represents the standard Brownian motion. Further, the distribution of $\mathbf{x}(t)$ is denoted as $p_t(\mathbf{x}(t))$, such that $\mathbf{x}(0) \sim p_0$, and $\mathbf{x}(T) \sim p_T$. $p_0$ is the data distribution, and $p_T$ is the prior distribution.

**Generativing samples by reversing the diffusion process.** By starting from sample $\mathbf{x}(T) \sim p_T$ and reversing the diffusion process, we can generate a sample from $p_0$. Since the reverse of a diffusion process is also a diffusion process [1], running backwards in time and given by the reverse-time SDE:

$$d\mathbf{x} = [\mathbf{f}(\mathbf{x}, t) - g(t)^2 \nabla_{\mathbf{x}} \log p_t(\mathbf{x})]dt + g(t)d\bar{\mathbf{w}}, \tag{2}$$

where $\bar{\mathbf{w}}$ is a Brownian motion in the reverse time direction. This reverse-time SDE can be computed once we know the drift and diffusion coefficients of Eq. (1), as well as the score for each $t \in [0, T]$.

**Estimating score via score matching.** From the above, it is known that the core of solving reverse-time SDE is to obtain the score for each time step $t$. The score can be estimated by training a time-dependent score network $s(\mathbf{x}(t), t; \theta)$ on training samples with the score matching [7, 24]:

$$\mathcal{L}_{\text{score}} = \mathbb{E}_{t \sim \mathcal{U}(0,T)}[\lambda(t)\mathbb{E}_{\mathbf{x}(0)}\mathbb{E}_{\mathbf{x}(t)|\mathbf{x}(0)}[\|s(\mathbf{x}(t), t; \theta) - \nabla_{\mathbf{x}(t)} \log p_{0t}(\mathbf{x}(t) \mid \mathbf{x}(0))\|_2^2]], \tag{3}$$

where $\mathcal{U}(0, T)$ is a uniform distribution over $[0, T]$, $p_{0t}(\mathbf{x}(t) \mid \mathbf{x}(0))$ denotes the transition kernel from $\mathbf{x}(0)$ to $\mathbf{x}(t)$, and $\lambda(t)$ denotes a positive weighting function. Finally, we need to know the transition kernel $p_{0t}(\mathbf{x}(t) \mid \mathbf{x}(0))$ to minimize Eq. (3) and optimize $s(\mathbf{x}(t), t; \theta)$. The transition kernel is always a Gaussian distribution when the drift coefficient of SDE is affine, where the mean and variance can be obtained with common techniques (see section 5.5 in [21]). For example, consider a SDE for score models: $d\mathbf{x} = \sigma^t d\mathbf{w}$, $t \in [0, 1]$, in this case, the corresponding transition kernel is

$$p_{0t}(\mathbf{x}(t) \mid \mathbf{x}(0)) = \mathcal{N}\left(\mathbf{x}(t); \mathbf{x}(0), \frac{1}{2\log \sigma}(\sigma^{2t} - 1)\mathbf{I}\right). \tag{4}$$

When we choose the weighting function $\lambda(t) = \frac{1}{2\log \sigma}(\sigma^{2t} - 1)$, the final objective $\mathcal{L}_{\text{score}}$ is

$$\mathcal{L}_{\text{score}} = \mathbb{E}_{\mathbf{x}, \mathbf{z} \sim \mathcal{N}(\mathbf{0}, \mathbf{I}), t \sim \mathcal{U}(0,T)}[\|\sqrt{\lambda(t)}s(\mathbf{x}(t), t; \theta) + \mathbf{z}\|_2^2], \tag{5}$$

where $\mathbf{z}$ is the random noise sampling from $\mathcal{N}(\mathbf{0}, \mathbf{I})$.

# 3 The Proposed Method

## 3.1 Problem Formulation

Let a tuple $(\mathbf{x}_1, \mathbf{x}_2, \cdots, \mathbf{x}_M)$ denotes $M$ heterogenous modalities of an example, where $\mathbf{x}_k$ is the input of the $k$-th modality. In the complete case, all modalities are observed and directly fused to facilitate MER. However, in many limited scenarios, there may be some modalities that are unavailable and need to be recovered for better fusion. For simplification, we introduce an indicator $\alpha \in \{0, 1\}$ to denote with $\alpha_k = 0$ if the $k$-th modality is missing, otherwise $\alpha_k = 1$. In the incomplete case, thus, the target is to recover those unobserved modalities $\mathcal{I}_{\text{miss}} = \{k|\alpha_k = 0\}$ for boosting the downstream tasks. It is worth noting that the missing emotion modalities may not be consistent for each example. To complete the missing modality $k \in \mathcal{I}_{\text{miss}}$, we could sample a large amount of data from its latent distribution $p(\mathbf{x}_k)$, as those advanced generative models do. But the sampling process does not have any semantic guidance, *how to generate semantically consistent samples in the sampling process (i.e., avoid semantic ambiguity)*? Fortunately, the observed modalities $\mathcal{I}_{\text{obs}} = \{k|\alpha_k = 1\}$ contain the semantic information we need, and thus can flavor the sampling process.

Our main idea is to recover the missing emotion modalities $\mathcal{I}_{\text{miss}}$ from their latent distribution spaces on the condition of the available modalities $\mathcal{I}_{\text{obs}}$. In other words, the observed emotion modalities $\mathcal{I}_{\text{obs}}$ are injected into the recovering process as the semantic conditions to constrain the distribution space of the sampling. Formally, we denote the data distribution of one missing modality as $p(\mathbf{x}_{\text{m}})$ ($\text{m} \in \mathcal{I}_{\text{miss}}$) and the available ones as $p(\mathbf{x}_{\mathcal{I}_{\text{obs}}})$, where $\mathbf{x}_{\mathcal{I}_{\text{obs}}}$ is abbreviated for brevity. Hereby, the ultimate goal is to sample missing modal data from the conditional distribution $p(\mathbf{x}_{\text{m}}|\mathbf{x}_{\mathcal{I}_{\text{obs}}})$. We consider a multi-step diffusion model to build this conditional distribution by perturbing $\mathbf{x}_{\text{m}}$, the perturbed conditional transition distribution is $p_t(\mathbf{x}_{\text{m}}(t)|\mathbf{x}_{\mathcal{I}_{\text{obs}}}(0))$ at the $t$-th step. Although $p_t(\mathbf{x}_{\text{m}}(t)|\mathbf{x}_{\mathcal{I}_{\text{obs}}}(0))$ is in general intractable, it can be approximated. We have

$$p_t(\mathbf{x}_{\text{m}}(t)|\mathbf{x}_{\mathcal{I}_{\text{obs}}}(0)) = \int p_t(\mathbf{x}_{\text{m}}(t)|\mathbf{x}_{\mathcal{I}_{\text{obs}}}(t), \mathbf{x}_{\mathcal{I}_{\text{obs}}}(0)) p_t(\mathbf{x}_{\mathcal{I}_{\text{obs}}}(t)|\mathbf{x}_{\mathcal{I}_{\text{obs}}}(0)) \, \mathrm{d}\mathbf{x}_{\mathcal{I}_{\text{obs}}}(t) \tag{6}$$

$$\approx \int p_t(\mathbf{x}_{\text{m}}(t)|\mathbf{x}_{\mathcal{I}_{\text{obs}}}(t)) p_t(\mathbf{x}_{\mathcal{I}_{\text{obs}}}(t)|\mathbf{x}_{\mathcal{I}_{\text{obs}}}(0)) \, \mathrm{d}\mathbf{x}_{\mathcal{I}_{\text{obs}}}(t) \tag{7}$$

where $p_t(\mathbf{x}_{\mathcal{I}_{\text{obs}}}(t)|\mathbf{x}_{\mathcal{I}_{\text{obs}}}(0))$ is tractable because it can be derived from the forward diffusion process. In the above approximation, we use the fact that $\mathbf{x}_{\mathcal{I}_{\text{obs}}}(t)$ is almost the same as $\mathbf{x}_{\mathcal{I}_{\text{obs}}}(0)$ when $t$ is small; if $t$ increases, $\mathbf{x}_{\mathcal{I}_{\text{obs}}}(0)$ becomes further away from $\mathbf{x}_{\text{m}}(t)$ in Markov chain so that $\mathbf{x}_{\text{m}}(t)$ is almost independent of $\mathbf{x}_{\mathcal{I}_{\text{obs}}}(0)$. Motivated by the score-based diffusion model [24], we compute the score of conditional transition probability in Eq. (6):

$$\nabla_{\mathbf{x}_{\text{m}}} \log p_t(\mathbf{x}_{\text{m}}(t)|\mathbf{x}_{\mathcal{I}_{\text{obs}}}(0)) \approx \nabla_{\mathbf{x}_{\text{m}}} \log \mathbb{E}_{p_t(\mathbf{x}_{\mathcal{I}_{\text{obs}}}(t)|\mathbf{x}_{\mathcal{I}_{\text{obs}}}(0))}[p_t(\mathbf{x}_{\text{m}}(t)|\mathbf{x}_{\mathcal{I}_{\text{obs}}}(t))] \tag{8}$$

$$\approx \nabla_{\mathbf{x}_{\text{m}}} \log p_t(\mathbf{x}_{\text{m}}(t)|\mathbf{x}_{\mathcal{I}_{\text{obs}}}(t)) \tag{9}$$

$$= \nabla_{\mathbf{x}_{\text{m}}} \log p_t([\mathbf{x}_{\text{m}}(t); \mathbf{x}_{\mathcal{I}_{\text{obs}}}(t)]), \tag{10}$$

where in the second step $\mathbf{x}_{\mathcal{I}_{\text{obs}}}(t)$ is a random sample from $p_t(\mathbf{x}_{\mathcal{I}_{\text{obs}}}(t)|\mathbf{x}_{\mathcal{I}_{\text{obs}}}(0))$, and note that $\nabla_{\mathbf{x}_{\text{m}}} \log p_t(\mathbf{x}_{\text{m}}(t)|\mathbf{x}_{\mathcal{I}_{\text{obs}}}(t)) = \nabla_{\mathbf{x}_{\text{m}}} \log p_t([\mathbf{x}_{\text{m}}(t); \mathbf{x}_{\mathcal{I}_{\text{obs}}}(t)])$ is held because:

$$\nabla_{\mathbf{x}_{\text{m}}} \log p_t([\mathbf{x}_{\text{m}}(t); \mathbf{x}_{\mathcal{I}_{\text{obs}}}(t)]) = \nabla_{\mathbf{x}_{\text{m}}} \log p_t(\mathbf{x}_{\text{m}}(t)|\mathbf{x}_{\mathcal{I}_{\text{obs}}}(t)) + \nabla_{\mathbf{x}_{\text{m}}} \log p_t(\mathbf{x}_{\mathcal{I}_{\text{obs}}}(t)) \tag{11}$$

$$= \nabla_{\mathbf{x}_{\text{m}}} \log p_t(\mathbf{x}_{\text{m}}(t)|\mathbf{x}_{\mathcal{I}_{\text{obs}}}(t)), \tag{12}$$

where $[\mathbf{x}_{\text{m}}(t); \mathbf{x}_{\mathcal{I}_{\text{obs}}}(t)]$ is a vector combination of $\mathbf{x}_{\text{m}}(t)$ and $\mathbf{x}_{\mathcal{I}_{\text{obs}}}(t)$ by a well-designed transformation function. This transformation function is called the *conditioning mechanism* that is described in Sec. 3.4. To infer the gradient, $\nabla_{\mathbf{x}_{\text{m}}} \log p_t([\mathbf{x}_{\text{m}}(t); \mathbf{x}_{\mathcal{I}_{\text{obs}}}(t)])$ could be parameterized a time-dependent conditioning score network $s_{\text{m}}(\mathbf{x}_{\text{m}}(t), \mathbf{x}_{\mathcal{I}_{\text{obs}}}(t), t; \theta_{\text{m}})$ for modality m. Finally, based on Eq. (5), we derive the conditional score matching objective to be optimized:

$$\mathcal{L}_{\text{score}}^{\text{cond}} = \mathbb{E}_{\mathbf{x}_{\text{m}}, \mathbf{x}_{\mathcal{I}_{\text{obs}}}, \mathbf{z} \sim \mathcal{N}(\mathbf{0}, \mathbf{I}), t \sim \mathcal{U}(0, T)}[\|\sqrt{\lambda(t)} s_{\text{m}}(\mathbf{x}_{\text{m}}(t), \mathbf{x}_{\mathcal{I}_{\text{obs}}}(t), t; \theta_{\text{m}}) + \mathbf{z}\|_2^2]. \tag{13}$$

Given sufficient data and model capacity, the well-trained $s_{\text{m}}(\mathbf{x}_{\text{m}}(t), \mathbf{x}_{\mathcal{I}_{\text{obs}}}(t), t; \theta_{\text{m}})$ can be used to solve the reverse-time SDE in Eq. (2) to generate each missing modality $\widetilde{\mathbf{x}}_{\text{m}}$, abided by $p(\mathbf{x}_{\text{m}})$.

Although the estimated $\widetilde{\mathbf{x}}_{\text{m}}$ obeys the original distribution, it would deviate from the ground truth when the scatter degree of intra-class samples is large. Thus, we can further refine it with a decoder $\mathcal{D}$, i.e., $\widehat{\mathbf{x}}_{\text{m}} = \mathcal{D}_{\text{m}}(\widetilde{\mathbf{x}}_{\text{m}})$. At the training stage, we minimize the reconstruction error between $\widehat{\mathbf{x}}_{\text{m}}$ and $\mathbf{x}_{\text{m}}$. And, we find this refinement step could enhance the performance as shown in Sec. 4.3.

## 3.2 Overview Framework

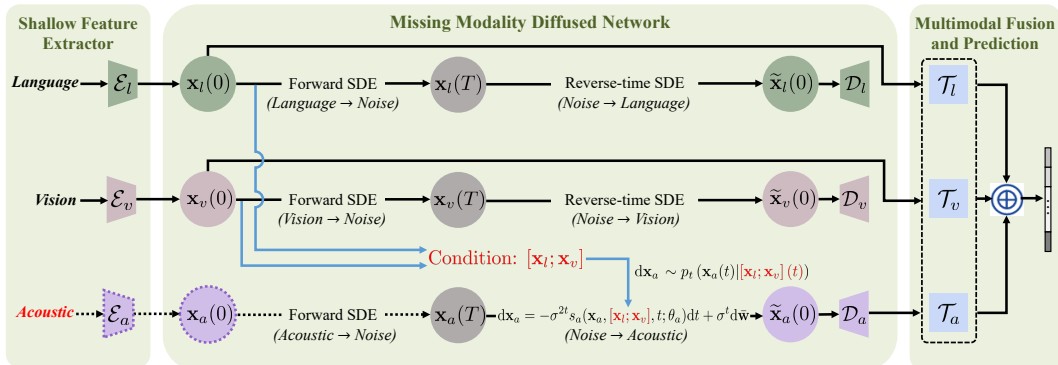

Figure 2: The framework of IMDer. Given the input incomplete data (acoustic modality is missed as an example), IMDer encodes shallow feature $\mathbf{x}_l$ and $\mathbf{x}_v$ (equal to $\mathbf{x}_l(0)$ and $\mathbf{x}_v(0)$) via shallow feature extractor (Sec. 3.3). In missing modality diffused network, we sample $\mathbf{x}_a(T)$ from prior noise distribution, and then denoise $\mathbf{x}_a(T)$ by solving the reverse-time diffusion with score model $s_a$ conditioned on $[\mathbf{x}_l; \mathbf{x}_v]$ to generate samples $\widetilde{\mathbf{x}}_a(0)$. Further, $\widetilde{\mathbf{x}}_a(0)$ is fed into the reconstruction module $\mathcal{D}_a$ to obtain the final recovered modality $\widehat{\mathbf{x}}_a$ (Sec. 3.4). Finally, $\widehat{\mathbf{x}}_a$ is combined with $\mathbf{x}_l$ and $\mathbf{x}_v$ as complete data for multimodal fusion and prediction to regress the emotional labels. (Sec. 3.5).

The overview framework is illustrated in Fig. 2. It mainly consists of three parts: **shallow feature extractor**, **missing modality diffused network**, and **multimodal fusion and prediction**. Due to the discrepancy of the original dimensional space between modalities, we first align the dimensionality of each modality by extracting multimodal shallow features to facilitate subsequent modality recovery. Then, to alleviate the distribution gap between recovered modalities and original ones, we design a missing modality diffused network to model the distribution of each missing modality and sample from the modeled distribution space to generate missing data. At the same time, to reduce the semantic ambiguity of the recovered modalities, the available modalities are embedded as conditions in the recovering process so that we can recover semantically consistent samples from a distribution conditioned on the available modalities. Finally, to perform the MER, the multimodal fusion and prediction part receives recovered complete multimodal data and uses multimodal transformers to fuse multimodal representation for emotion recognition. The detail is introduced in the next subsections.

### 3.3 Shallow Feature Extractor

We consider three heterogeneous modalities: *language* ($l$), *vision* ($v$), and *acoustic* ($a$) for the MER task. Since the original dimensional spaces of the three modalities are often inconsistent, they are not suitable for direct use in missing modality recovering. To address this problem, we design a shallow feature extractor that contains three independent encoders $\mathcal{E}_k$, $k \in \{l, v, a\}$, to extract the shallow features of three modalities and map them into the same dimensional space. Each encoder contains a 1D convolutional layer. Thus, the subsequent recovery task aims to recover missing modalities from the distribution space of shallow features.

Given an input example (aka sequence data), we can extract the shallow features, $\mathcal{X} = \{\mathbf{x}_k\}, \mathbf{x}_k \in \mathbb{R}^{L \times D}$, $L$ and $D$ indicate the sequence length and the feature dimensionality. In the incomplete multimodal case, some modalities are missing either fixedly or randomly by guaranteeing at least one modality is available in $\mathcal{X}$. For the three modalities mentioned above, a total of seven missing combinational cases are included, which are reported in Tab. 1. For a convenient statement but without loss of generality, below we recover the missed acoustic modality $\mathbf{x}_a$, as shown in Fig. 2.

### 3.4 Missing Modality Diffused Network

We consider a score network $s_{\mathrm{m}}$ to model the distribution of missing modality $\mathrm{m} \in \mathcal{I}_{\mathrm{miss}}$ by perturbing $\mathbf{x}_{\mathrm{m}}$ with a common SDE: $\mathrm{d}\mathbf{x} = \sigma^t \mathrm{d}\mathbf{w}$, $t \in [0, 1]$, like score-based diffusion [24], the corresponding reverse-time SDE can be derived from Eq. (2) and Eq. (8) as:

$$\mathrm{d}\mathbf{x}_{\mathrm{m}} = -\sigma^{2t} s_{\mathrm{m}}(\mathbf{x}_{\mathrm{m}}, \mathbf{x}_{\mathcal{I}_{\mathrm{obs}}}, t; \theta_{\mathrm{m}})\mathrm{d}t + \sigma^t \mathrm{d}\bar{\mathbf{w}}, \tag{14}$$

where $\mathbf{x}_{\mathcal{I}_{\text{obs}}}$ is a condition combining all available modalities. When the score network is well-trained based on Eq. (13), we can generate missing data via Eq. (14). Taking the example that the language modality $\mathbf{x}_l$ and vision modality $\mathbf{x}_v$ are available, the acoustic modality $\mathbf{x}_a$ is missing, we build a score network $s_a$ conditioned on $\mathbf{x}_l$ and $\mathbf{x}_v$ to recover $\mathbf{x}_a$ with the following reverse-time SDE:

$$\mathrm{d}\mathbf{x}_a = -\sigma^{2t} s_a(\mathbf{x}_a, [\mathbf{x}_l; \mathbf{x}_v], t; \theta_a)\mathrm{d}t + \sigma^t \mathrm{d}\bar{\mathbf{w}}, \tag{15}$$

where $[\mathbf{x}_l; \mathbf{x}_v]$ denotes a fused feature that is combined of $\mathbf{x}_l$ and $\mathbf{x}_v$ by a feature concatenation operation and a 1D convolution layer with a kernel size of 1. Note that the above feature fusion is not required when only one modality is available. To recover the missing modalities using the above equation, we need to solve the reverse-time SDE with numerical methods. Here, we use a general-purpose Euler–Maruyama numerical method that is based on a simple discretization to the SDE [24]. When applied to Eq. (15), we can obtain the following iteration rule

$$\mathbf{x}_a(t - \Delta t) = \mathbf{x}_a(t) + \sigma^{2t} s_a(\mathbf{x}_a(t), [\mathbf{x}_l; \mathbf{x}_v](t), t; \theta_a)\Delta t + \sigma^t \sqrt{\Delta t}\epsilon_t, \tag{16}$$

where $\Delta t$ is a discrete time step size and $\epsilon_t \sim \mathcal{N}(\mathbf{0}, \mathbf{I})$. Given sufficient iteration steps, we can obtain recovered data $\widetilde{\mathbf{x}}_a$. Subsequently, $\widetilde{\mathbf{x}}_a$ is fed into the reconstruction module to obtain the refined acoustic modality: $\widehat{\mathbf{x}}_a = \mathcal{D}_a(\widetilde{\mathbf{x}}_a)$, where $\mathcal{D}_a$ denotes the reconstruction module of acoustic modality. For any missing patterns, the set of recovered features can be denoted as $\widehat{\mathcal{X}}_{\text{miss}} = \{\widehat{\mathbf{x}}_k | k \in \mathcal{I}_{\text{miss}}\}$, thus the reconstruction loss $\mathcal{L}_{\text{rec}}$ is denoted as:

$$\mathcal{L}_{\text{rec}} = \sum_{k \in \mathcal{I}_{\text{miss}}} \|\widehat{\mathbf{x}}_k - \mathbf{x}_k\|_2^2. \tag{17}$$

Further, to optimize the score networks, we can obtain the score matching objective $\mathcal{L}_{\text{score}}$ under any missing patterns according to Eq. (13):

$$\mathcal{L}_{\text{score}} = \sum_{k \in \mathcal{I}_{\text{miss}}} \mathbb{E}_{\mathbf{x}_k, \mathbf{x}_{\mathcal{I}_{\text{obs}}}, \mathbf{z} \sim \mathcal{N}(\mathbf{0}, \mathbf{I}), t \sim \mathcal{U}(0, T)} [\|\sqrt{\lambda(t)} s_k(\mathbf{x}_k(t), \mathbf{x}_{\mathcal{I}_{\text{obs}}}(t), t; \theta_k) + \mathbf{z}\|_2^2]. \tag{18}$$

By combining $\mathcal{L}_{\text{rec}}$ and $\mathcal{L}_{\text{score}}$, the loss of missing modality diffused network is $\mathcal{L}_{\text{mmd}} = \mathcal{L}_{\text{rec}} + \mathcal{L}_{\text{score}}$.

**Conditioning mechanism.** Besides the above, another vital question is how to embed the available modalities (i.e., conditions) into the sample generation. Inspired by cross-modal attention mechanism [27], we exploit this mechanism to fuse intermediate representation of $\mathbf{x}_k(t), k \in \mathcal{I}_{\text{miss}}$ and $\mathbf{x}_{\mathcal{I}_{\text{obs}}}(t)$ in $s_k$ to achieve condition embedding with a flexible manner. In particular, we assume that a certain intermediate representation of $\mathbf{x}_k(t)$ and $\mathbf{x}_{\mathcal{I}_{\text{obs}}}(t)$ in $s_k$ is $\mathbf{x}_k^{\text{ir}}(t)$ and $\mathbf{x}_{\mathcal{I}_{\text{obs}}}^{\text{ir}}(t)$, where $\mathbf{x}_k^{\text{ir}}(t)$ and $\mathbf{x}_{\mathcal{I}_{\text{obs}}}^{\text{ir}}(t)$ have same tensor size. The cross-modal attention mechanism can be defined as:

$$\mathbf{x}_{\text{cond}}^{\text{ir}}(t) = \text{softmax}(\frac{\mathbf{Q}\mathbf{K}^\top}{\sqrt{d}})\mathbf{V}, \tag{19}$$

where $\mathbf{x}_{\text{cond}}^{\text{ir}}(t)$ denotes the condition embedded representation, $d$ denotes dimensionality of intermediate representation. $\mathbf{Q} = \mathbf{x}_k^{\text{ir}}(t)\mathbf{W}_{\mathbf{Q}}$, $\mathbf{K} = \mathbf{x}_{\mathcal{I}_{\text{obs}}}^{\text{ir}}(t)\mathbf{W}_{\mathbf{K}}$, and $\mathbf{V} = \mathbf{x}_{\mathcal{I}_{\text{obs}}}^{\text{ir}}(t)\mathbf{W}_{\mathbf{V}}$, where $\mathbf{W}_{\mathbf{Q}}, \mathbf{W}_{\mathbf{K}}$ and $\mathbf{W}_{\mathbf{V}}$ are the learnable parameters. Since we consider a UNet-style architecture to build the score network, we apply Eq. (19) for condition embedding at each different size of intermediate representation. More details of the score network are elaborated in the supplementary file.

## 3.5 Multimodal Fusion and Prediction

The recovered data $\widehat{\mathcal{X}}_{\text{miss}} = \{\widehat{\mathbf{x}}_k | k \in \mathcal{I}_{\text{miss}}\}$ and the available data $\mathcal{X}_{\text{obs}} = \{\mathbf{x}_k | k \in \mathcal{I}_{\text{obs}}\}$ are combined as the complete multimodal data for multimodal fusion and prediction of emotions. We employ multimodal transformers $\mathcal{T}_k$ [27] to fuse complete multimodal data $\widehat{\mathcal{X}}_{\text{miss}} \cup \mathcal{X}_{\text{obs}}$, and the fused feature is used to predict emotions by fully connected layers. We integrate the above losses to reach the full optimization objective:

$$\mathcal{L}_{\text{total}} = \mathcal{L}_{\text{task}} + \beta \mathcal{L}_{\text{mmd}}, \tag{20}$$

where $\mathcal{L}_{\text{task}}$ is the task loss defined as mean absolute error, $\beta$ controls the importance of different loss functions. The entire optimization is implemented in an end-to-end manner, and the concrete training about the configuration of incomplete modalities could be found in the experiment part.

# 4 Experiments

## 4.1 Datasets and Implementation Details

**Datasets.** We consider two standard MER datasets to conduct experiments, including CMU-MOSI [32] and CMU-MOSEI [33]. CMU-MOSI consists of 2199 monologue video clips. Where 1284, 229, and 686 samples are used as training, validation, and testing set. CMU-MOSEI contains 22856 samples of movie review video clips. Where 16326 samples are used for training, the remaining 1871 and 4659 samples are used for validation and testing. On the two datasets, we extract the language features via pre-trained BERT model [9] and obtain a 768-dimensional hidden state as the word embedding. For vision modality, each video frame was encoded via Facet [8] to represent the presence of the total 35 facial action units. The acoustic modality was processed by COVAREP [5] to obtain the 74-dimensional features.

**Evaluation metrics.** CMU-MOSI and CMU-MOSEI are collections of videos from YouTube of people providing opinions, that have different human annotations, including human emotion score and emotion categories (only for CMU-MOSEI). Our work focuses on human emotion score. Each sample in two datasets was labeled with a human emotion score that ranges from -3 to 3, including *highly negative*, *negative*, *weakly negative*, *neutral*, *weakly positive*, *positive*, and *highly positive*. To make a comprehensive comparison, we evaluate the performance using the following metrics: 7-class accuracy ($ACC_7$), binary accuracy ($ACC_2$), and F1 score.

**Implementation details.** Following [20, 35, 14], we investigate the performance of different methods under two commonly-used protocols: (1) *fixed missing protocol*. (2) *random missing protocol*. For the *fixed missing protocol*, we consistently discard one modality (i.e., $\{l, v\}$, $\{l, a\}$, $\{v, a\}$) or two modalities (i.e., $\{l\}$, $\{v\}$, $\{a\}$). For the *random missing protocol*, the missing patterns are randomized for each sample (i.e., one or two modalities may be missing for each sample). Here, we use the missing rate (MR) to measure the overall missingness of the dataset. The MR is defined as $MR = 1 - \frac{\sum_{i=1}^{N} m_i}{N \times M}$, where $m_i$ denotes the number of available modalities for $i^{th}$ sample, $N$ denotes the total number of samples, and $M$ indicates the number of modalities. We also ensure that at least one modality is available for each sample, so $m_i \geq 1$ and $MR \leq \frac{M-1}{M}$. For three modalities, we choose the MR from $[0.0, 0.1, \cdots, 0.7]$, where 0.7 is an approximation of $\frac{M-1}{M}$ with the same meaning. We keep the same MR during training, validation, and testing phases, consistent with previous work [14]. The optimal setting for $\beta$ is set to 0.1 via the performance on the validation set. We run each experiment five times and report the average values on the testing set.

Table 1: MER accuracy comparison under *fixed missing protocol*. The values reported in each cell denote $ACC_2$/F1/$ACC_7$. **Bold** is the best.

| Datasets | Available | DCCA [2] | DCCAE [29] | MCTN [20] | MMIN [35] | GCNet [14] | **IMDer (Ours)** |
|---|---|---|---|---|---|---|---|
| MOSI | $\{l\}$ | 73.6 / 73.8 / 30.2 | 76.4 / 76.5 / 28.3 | 79.1 / 79.2 / 41.0 | 83.8 / 83.8 / 41.6 | 83.7 / 83.6 / 42.3 | **84.8 / 84.7 / 44.8** |
| | $\{v\}$ | 47.7 / 41.5 / 16.6 | 52.6 / 51.1 / 17.1 | 55.0 / 54.4 / 16.3 | 57.0 / 54.0 / 15.5 | 56.1 / 55.7 / 16.9 | **61.3 / 60.8 / 22.2** |
| | $\{a\}$ | 50.5 / 46.1 / 16.3 | 48.8 / 42.1 / 16.9 | 56.1 / 54.5 / 16.5 | 55.3 / 51.5 / 15.5 | 56.1 / 54.5 / 16.6 | **62.0 / 62.2 / 22.0** |
| | $\{l, v\}$ | 74.9 / 75.0 / 30.3 | 76.7 / 76.8 / 30.0 | 81.1 / 81.2 / 42.1 | 83.8 / 83.9 / 42.0 | 84.3 / 84.2 / 43.4 | **85.5 / 85.4 / 45.3** |
| | $\{l, a\}$ | 74.7 / 74.8 / 29.7 | 77.0 / 77.0 / 30.2 | 81.0 / 81.0 / 43.2 | 84.0 / 84.0 / 42.3 | 84.5 / 84.4 / 43.4 | **85.4 / 85.3 / 45.0** |
| | $\{v, a\}$ | 50.8 / 46.4 / 16.6 | 54.0 / 52.5 / 17.4 | 57.5 / 57.4 / 16.8 | 60.4 / 58.5 / 19.5 | 62.0 / 61.9 / 17.2 | **63.6 / 63.4 / 23.8** |
| | $\{l, v, a\}$ | 75.3 / 75.4 / 30.5 | 77.3 / 77.4 / 31.2 | 81.4 / 81.5 / 43.4 | 84.6 / 84.4 / 44.8 | 85.2 / 85.1 / 44.9 | **85.7 / 85.6 / 45.3** |
| | Average | 63.9 / 61.9 / 20.0 | 66.1 / 64.8 / 24.4 | 70.2 / 69.9 / 31.3 | 72.7 / 71.4 / 31.6 | 73.1 / 72.8 / 32.1 | **75.5 / 75.3 / 35.5** |
| MOSEI | $\{l\}$ | 78.5 / 78.7 / 46.7 | 79.7 / 79.5 / 47.0 | 82.6 / 82.8 / 50.2 | 82.3 / 82.4 / 51.4 | 83.0 / 83.2 / 51.2 | **84.5 / 84.5 / 52.5** |
| | $\{v\}$ | 61.9 / 55.7 / 41.3 | 61.1 / 57.2 / 40.1 | 62.6 / 57.1 / 41.6 | 59.3 / 60.0 / 40.7 | 61.9 / 61.6 / 41.7 | **63.9 / 63.6 / 42.6** |
| | $\{a\}$ | 62.0 / 50.2 / 41.1 | 61.4 / 53.8 / 40.9 | 62.7 / 54.5 / 41.4 | 58.9 / 59.5 / 40.4 | 60.2 / 60.3 / 41.1 | **63.8 / 60.6 / 41.7** |
| | $\{l, v\}$ | 80.3 / 79.7 / 46.6 | 80.4 / 80.4 / 47.1 | 83.2 / 83.2 / 50.4 | 83.8 / 83.4 / 51.2 | 84.3 / 84.4 / 51.1 | **85.0 / 85.0 / 53.1** |
| | $\{l, a\}$ | 79.5 / 79.2 / 46.7 | 80.0 / 80.0 / 47.4 | 83.5 / 83.3 / 50.7 | 83.7 / 83.3 / 52.0 | 84.3 / 84.4 / 51.3 | **85.1 / 85.1 / 53.1** |
| | $\{v, a\}$ | 63.4 / 56.9 / 41.5 | 62.7 / 59.2 / 41.6 | 63.7 / 62.7 / 42.1 | 63.5 / 61.9 / 41.8 | 64.1 / 57.2 / 42.0 | **64.9 / 63.5 / 42.8** |
| | $\{l, v, a\}$ | 80.7 / 80.9 / 47.7 | 81.2 / 81.2 / 48.2 | 84.2 / 84.2 / 51.2 | 84.3 / 84.2 / 52.4 | **85.2 / 85.1** / 51.5 | 85.1 / **85.1 / 53.4** |
| | Average | 72.3 / 68.8 / 44.5 | 72.4 / 70.2 / 44.6 | 74.6 / 72.5 / 46.8 | 73.7 / 73.5 / 47.1 | 74.7 / 73.7 / 47.1 | **76.0 / 75.3 / 48.5** |

## 4.2 Comparison with the state-of-the-arts

We compare IMDer with the current state-of-the-art incomplete multimodal emotion recognition methods based on modality recovery mechanism, including MCTN [20], MMIN [35], GCNet [14]. For a comprehensive comparison, we consider two deep learning-based non-recovery methods with

canonical correlation maximization DCCA [2] and DCCAE [29] in incomplete multimodal learning to conduct comparisons. Below, we report the quantitative and qualitative experimental results.

Table 2: MER accuracy comparison under *random missing protocol*. The values reported in each cell denote $ACC_2$/F1/$ACC_7$. MR: Missing Rate. **Bold** is the best.

| Datasets | MR | DCCA [2] | DCCAE [29] | MCTN [20] | MMIN [35] | GCNet [14] | **IMDer (Ours)** |
|---|---|---|---|---|---|---|---|
| MOSI | 0.0 | 75.3 / 75.4 / 30.5 | 77.3 / 77.4 / 31.2 | 81.4 / 81.5 / 43.4 | 84.6 / 84.4 / 44.8 | 85.2 / 85.1 / 44.9 | **85.7 / 85.6 / 45.3** |
| | 0.1 | 72.1 / 72.2 / 28.0 | 74.5 / 74.7 / 28.1 | 78.4 / 78.5 / 39.8 | 81.8 / 81.8 / 41.2 | 82.3 / 82.3 / 42.1 | **84.9 / 84.8 / 44.8** |
| | 0.2 | 69.3 / 69.1 / 26.8 | 71.8 / 71.9 / 27.6 | 75.6 / 75.7 / 38.5 | 79.0 / 79.1 / 38.9 | 79.4 / 79.5 / 40.0 | **83.5 / 83.4 / 44.3** |
| | 0.3 | 65.4 / 65.2 / 25.7 | 67.0 / 66.7 / 25.8 | 71.3 / 71.2 / 35.5 | 76.1 / 76.2 / 36.9 | 77.2 / 77.2 / 38.2 | **81.2 / 81.0 / 42.5** |
| | 0.4 | 62.8 / 62.0 / 24.2 | 63.6 / 62.8 / 24.2 | 68.0 / 67.6 / 32.9 | 71.7 / 71.6 / 34.9 | 74.3 / 74.4 / 36.6 | **78.6 / 78.5 / 39.7** |
| | 0.5 | 60.9 / 59.9 / 21.6 | 62.0 / 61.3 / 23.0 | 65.4 / 64.8 / 31.2 | 67.2 / 66.5 / 32.2 | 70.0 / 69.8 / 33.9 | **76.2 / 75.9 / 37.9** |
| | 0.6 | 58.6 / 57.3 / 21.2 | 59.6 / 58.5 / 20.9 | 63.8 / 62.5 / 29.7 | 64.9 / 64.0 / 29.1 | 67.7 / 66.7 / 29.8 | **74.7 / 74.0 / 35.8** |
| | 0.7 | 57.4 / 56.0 / 20.4 | 58.1 / 57.4 / 20.6 | 61.2 / 59.0 / 27.5 | 62.8 / 61.0 / 28.4 | 65.7 / 65.4 / 28.1 | **71.9 / 71.2 / 33.4** |
| | Average | 65.2 / 64.6 / 24.8 | 66.7 / 66.3 / 25.2 | 70.6 / 70.1 / 34.8 | 73.5 / 73.1 / 35.8 | 75.2 / 75.1 / 36.7 | **79.6 / 79.3 / 40.5** |
| MOSEI | 0.0 | 80.7 / 80.9 / 47.7 | 81.2 / 81.2 / 48.2 | 84.2 / 84.2 / 51.2 | 84.3 / 84.2 / 52.4 | 85.2 / 85.1 / 51.5 | **85.1 / 85.1 / 53.4** |
| | 0.1 | 77.4 / 77.3 / 46.2 | 78.4 / 78.3 / 46.9 | 81.8 / 81.6 / 49.8 | 81.9 / 81.3 / 50.6 | 82.3 / 82.1 / 51.2 | **84.8 / 84.6 / 53.1** |
| | 0.2 | 73.8 / 74.0 / 45.1 | 75.5 / 75.4 / 46.3 | 79.0 / 78.7 / 48.6 | 79.8 / 78.8 / 49.6 | 80.3 / 79.9 / 50.2 | **82.7 / 82.4 / 52.0** |
| | 0.3 | 71.1 / 71.2 / 43.6 | 72.3 / 72.2 / 45.6 | 76.9 / 76.2 / 47.4 | 77.2 / 75.5 / 48.1 | 77.5 / 76.8 / 49.2 | **81.3 / 80.7 / 51.3** |
| | 0.4 | 69.5 / 69.4 / 43.1 | 70.3 / 70.0 / 44.0 | 74.3 / 74.1 / 45.6 | 75.2 / 72.6 / 47.5 | 76.0 / 74.9 / 48.0 | **79.3 / 78.1 / 50.0** |
| | 0.5 | 67.5 / 65.4 / 42.5 | 69.2 / 66.4 / 43.3 | 73.6 / 72.6 / 45.1 | 73.9 / 70.7 / 46.7 | 74.9 / 73.2 / 46.7 | **79.0 / 77.4 / 49.2** |
| | 0.6 | 66.2 / 63.1 / 42.4 | 67.6 / 63.2 / 42.9 | 73.2 / 71.1 / 43.8 | 73.2 / 70.3 / 45.6 | 74.1 / 72.1 / 45.1 | **78.0 / 75.5 / 48.5** |
| | 0.7 | 65.6 / 61.0 / 42.1 | 66.6 / 62.6 / 42.5 | 72.7 / 70.5 / 43.6 | 73.1 / 69.5 / 44.8 | 73.2 / 70.4 / 44.5 | **77.3 / 74.6 / 47.6** |
| | Average | 71.5 / 70.3 / 44.1 | 72.6 / 71.2 / 45.0 | 77.0 / 76.1 / 46.9 | 77.3 / 75.4 / 48.2 | 77.9 / 76.8 / 48.3 | **80.9 / 79.8 / 50.6** |

**Quantitative results.** Tab. 1 and Tab. 2 list the quantitative results of fixed missing protocol and random missing protocol on two datasets, respectively. We have the following observations:

**1.** IMDer achieves the best results on the two MER datasets under both *fixed missing protocol* and *random missing protocol*, demonstrating superiority of IMDer. Compared with the non-recovery-based methods [2, 29], IMDer obtains consistent MER performance gains. This can be explained that IMDer can explicitly recover the missing modalities which can provide extra complementary information for MER. Compared with the recovery-based methods [20, 35, 14], the consistent improvements of IMDer indicate the superiority of maintaining distribution consistency between the recovered and the original modalities.

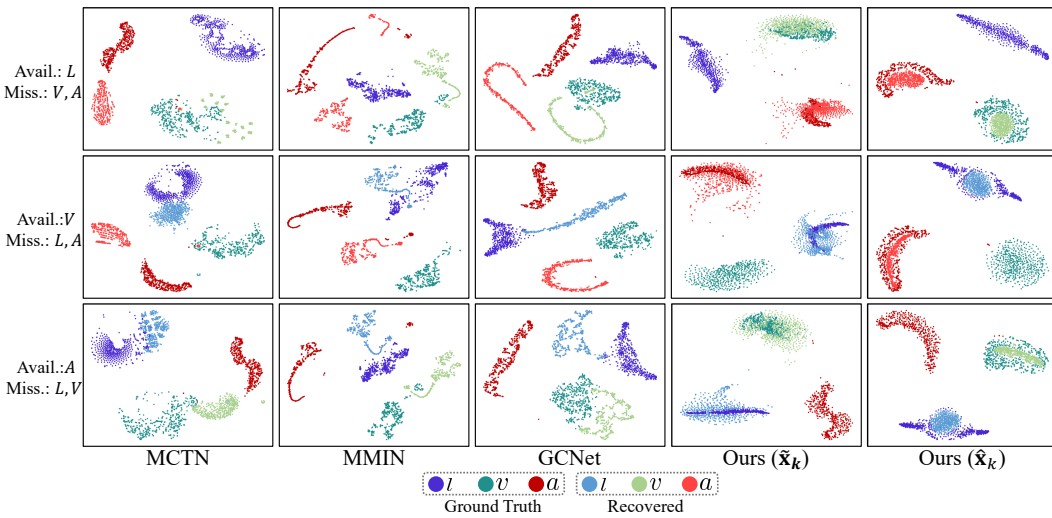

Figure 3: Visualization of recovered modalities. *Avail.* means available. The distribution of modality recovered by our method is much closer to the ground truth than the compared methods.

**2.** From the experimental results under *fixed missing protocol* in Tab. 1, we can observe that when *language* modality is missing, our proposed IMDer obtains more obvious enhancements. This is reasonable because language modality is crucial for robust MER, i.e., it is highly abstract and semantic and contains more discriminative information [20]. Thus, it is vital to complete the missing language modality for robust MER. Compared with other recovery-based methods, our proposed IMDer explicitly models the distribution space of the language modality and recovers the distributionally and

semantically consistent language features to facilitate robust MER. The comprehensive comparison in Tab. 1 verify the effectiveness of our method.

**3.** Compared with other MER methods, IMDer suffers from less performance degradation as the modality missing rate increases. This can be verified in Tab. 2 (*random missing protocol*). On CMU-MOSI, when the missing rate increases from 0.0 to 0.7, the $ACC_2$ of the compared recovery-based methods declines $19.5\% \sim 21.8\%$ while our IMDer merely declines 13.8%. For the CMU-MOSEI, as the missing rate increases from 0.0 to 0.7, $ACC_2$ of other recovery-based methods show obvious decline: $11.2\% \sim 12\%$. As a comparison, our proposed IMDer shows 7.8% accuracy degradation. These comparisons demonstrate IMDer's generalizability under random missing protocol.

**Qualitative results.** Fig. 3 visualizes the distribution of recovered data and ground truth for different recovery-based methods under the *fixed missing protocol*. We randomly select 500 samples in the testing set from the CMU-MOSEI dataset. The features of the selected samples are projected into a 2D space by t-SNE [28]. From these results, we can observe that the distribution between the original and recovered modalities estimated by IMDer is much closer than other methods, indicating IMDer effectively mitigates the distribution gap between recovered and the original data.

Table 3: Ablation study of the key components in IMDer under the average random missing protocol.

| Datasets | $s_k$ | $\mathcal{D}_k$ | $ACC_2$ | F1 | $ACC_7$ |
|---|---|---|---|---|---|
| CMU-MOSI | ✓ | ✓ | **79.6** | **79.3** | **40.5** |
| | ✓ | × | 77.8 | 77.7 | 38.4 |
| | × | ✓ | 75.8 | 75.6 | 36.8 |
| | × | × | 74.5 | 74.4 | 36.0 |
| CMU-MOSEI | ✓ | ✓ | **80.9** | **79.8** | **50.6** |
| | ✓ | × | 79.8 | 78.5 | 49.4 |
| | × | ✓ | 78.0 | 76.9 | 48.8 |
| | × | × | 76.5 | 76.0 | 48.5 |

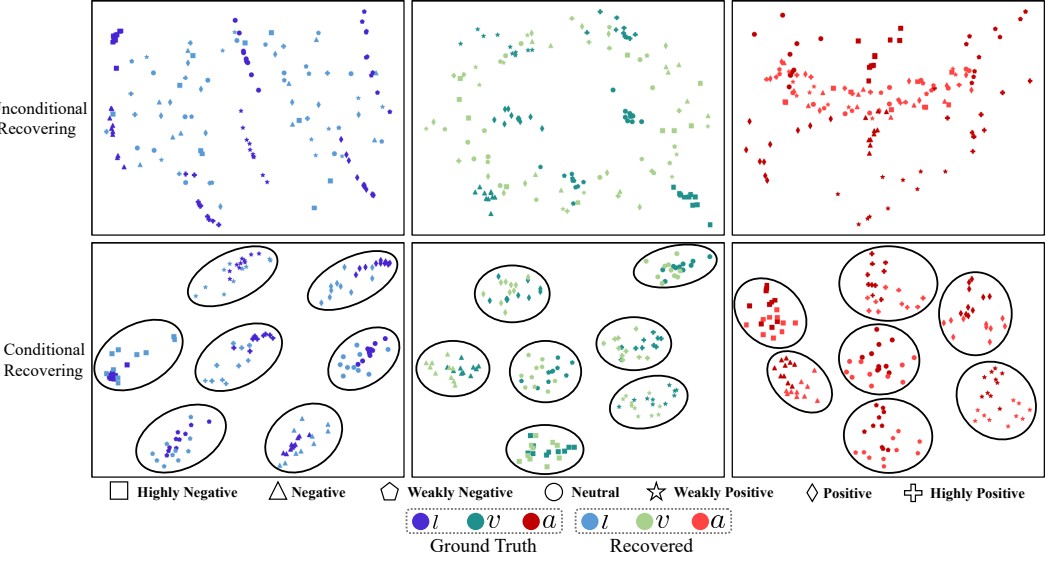

Figure 4: Visualization of the unconditional and conditional modality recovering. Conditional recovering (bottom row) shows the promising emotion category separability, indicating that using the vanilla available modalities as conditions can reduce the semantic ambiguity.

### 4.3 Ablation study

**Quantitative analysis.** We evaluate the effects of the key components in IMDer, including score model $s_k$ and reconstruction module $\mathcal{D}_k$. The results are illustrated in Tab. 3, we conclude the conclusions as: **1)** Recovering missing modalities by reconstruction module $\mathcal{D}_k$ or score model $s_k$ is effective. It is because explicitly recovering the modality can provide meaningful complementary

information. Further, we find that the score model-based modality recovery can provide more performance improvements. This is because the score model can recover data from the distribution of each missing modality, and the distribution between recovered data and original data is more consistent. **2)** Combing $s_k$ with $\mathcal{D}_k$ brings further benefits, which proves the refinement mechanism provided by $\mathcal{D}_k$ and the diffusion-based score model $s_k$ can benefit each other mutually.

**Visualization of unconditional/conditional recovering.** Fig. 4 visualizes the recovered and the original modality under the unconditional and conditional settings. Specifically, we randomly select 70 samples (ten samples for each class) in the testing set of the CMU-MOSEI dataset. The features of the selected samples are projected into a 2D space by t-SNE [28]. We can observe that unconditional recovering suffer from semantic ambiguity due to the lack of semantic guidance. In contrast, the data recovered by conditional recovering has been clustered and can be distinguished across seven different emotion classes, indicating that conditional recovering can mitigate semantic ambiguity between the recovered and the original modalities.

## 5 Conclusion and Discussion

In this paper, we try to challenge MER under incomplete multimodalities by proposing an Incomplete Multimodality-Diffused emotion recognition (IMDer) method. To recover the missing modalities, IMDer exploits the score-based diffusion model that maps the input random noise into the distribution space of missing modalities and recovers missing data abided by their original distributions. To reduce semantic ambiguity between the missing and the recovered modalities, the available modalities are used as conditions to guide the diffusion-based recovering process. Quantitative and qualitative experiments consistently demonstrate the effectiveness of IMDer.

**Limitations.** This work also has some limitations. The first limitation is that solving the reverse-time SDE incurs high computational overhead and may not be suitable for scenarios requiring real-time performance, and we will try to mitigate this issue in future work, such as replacing the Euler–Maruyama solvers with ordinary differential equation solvers. In addition, our work may not work to recognize emotion in other scenarios such as human-robot interaction, because emotion signal might change depending on the context, culture, etc.

**Ethical implications discussion.** Recognizing human emotion from multimodalities helps build better human-computer interactions. Albeit the obvious benefits, this technology also introduces some potential negative societal impact. For example, multimodal data contains a large number of faces, facial data will bring potential social privacy and security issues.

## 6 Acknowledgement

This work was supported by the National Natural Science Foundation of China (Grant Nos. 62072244, 62102180), the fundamental research funds for the central universities (Grant No. 30919011232), the Natural Science Foundation of Shandong Province (Grant Nos. ZR2020LZH008, ZR2022LZH003), the Natural Science Foundation of Jiangsu Province (Grant No. BK20210329), Shuangchuang Program of Jiangsu Province(Grant No. JSSCBS20210210), and in part by State Key Laboratory of High-end Server & Storage Technology.

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
