# Supplementary Materials for
## *Incomplete Multimodality-Diffused Emotion Recognition*

**Yuanzhi Wang, Yong Li, Zhen Cui**

PCA Lab, Key Lab of Intelligent Perception and Systems for High-Dimensional
Information of Ministry of Education, School of Computer Science and Engineering,
Nanjing University of Science and Technology, Nanjing, China.
{yuanzhiwang, yong.li, zhen.cui}@njust.edu.cn

## 1 Overview

In this supplementary material, we first present the details of the conditional score network in Sec. 2.
Then, we provide the detailed neural network configurations as well as the hyper-parameter settings
in Sec. 3. Next, we report the quantitative results of both unconditional and conditional recovering in
Sec. 4. Finally, we conduct experiments on Chinese MER dataset CH-SIMS [6] for further discussion
and comparison in Sec. 5.

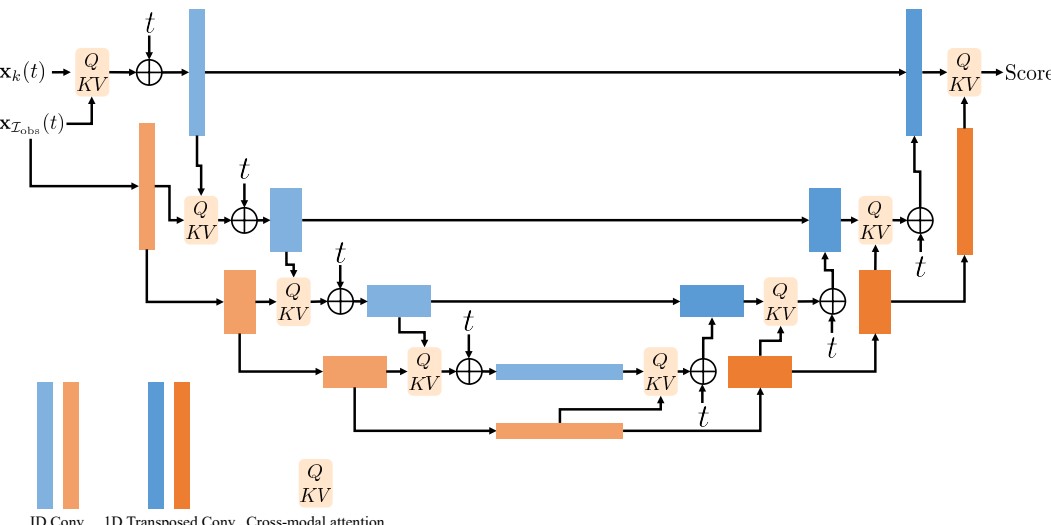

Figure 1: Model architecture of the conditional score network.

## 2 Conditional Score Network

We exploit a conditional score network $s_k$ to estimate the score of the missing modality $k \in \mathcal{I}_{\mathrm{miss}}$ at
each time step $t \in [0, 1]$, where the input of $s_k$ consists of perturbed missing data $\mathbf{x}_k(t)$, perturbed

---

Corresponding authors: Yong Li, Zhen Cui

37th Conference on Neural Information Processing Systems (NeurIPS 2023).

available data $\mathbf{x}_{\mathcal{I}_{\mathrm{obs}}}(t)$, and the corresponding temporal information $t$. When multiple available modalities exist, all of them will be concatenated and fused by a 1D convolutional layer with a kernel size of 1. As shown in Fig. 1, we use the UNet-style neural network as the backbone of $s_k$, where we use the 1D *vanilla* and *transposed* convolutional layers to build the UNet backbone, aiming to spot the sequential multimodal features. For condition embedding, we exploit the cross-modal attention mechanism [5]. Typically, we inject the available modalities into each intermediate feature within the UNet. Let us take the $i$-th intermediate representation $\mathbf{x}_k^i(t)$ and $\mathbf{x}_{\mathcal{I}_{\mathrm{obs}}}^i(t)$ of $\mathbf{x}_k(t)$ and $\mathbf{x}_{\mathcal{I}_{\mathrm{obs}}}(t)$ in $s_k$ as an example, the cross-modal attention based condition embedding can be formulated as:

$$\mathbf{x}_{\mathrm{cond}}^i(t) = \mathrm{softmax}(\frac{\mathbf{Q}\mathbf{K}^\top}{\sqrt{d}})\mathbf{V}, \tag{1}$$

where $\mathbf{x}_{\mathrm{cond}}^i(t)$ denotes the embedded condition representation which will be used in the next step. $d$ denotes the dimensionality of intermediate representation. Furthermore, $\mathbf{Q} = \mathbf{x}_k^i(t)\mathbf{W_Q}$, $\mathbf{K} = \mathbf{x}_{\mathcal{I}_{\mathrm{obs}}}^i(t)\mathbf{W_K}$ and $\mathbf{V} = \mathbf{x}_{\mathcal{I}_{\mathrm{obs}}}^i(t)\mathbf{W_V}$. $\mathbf{W_Q}, \mathbf{W_K}, \mathbf{W_V}$ are the learnable parameters. In addition, we incorporate the temporal information $t$ via gaussian random features [4]. Specifically, we first sample $\omega \sim \mathcal{N}(\mathbf{0}, s^2\mathbf{I})$ which is subsequently fixed for the model (i.e., not learnable). For a time step $t$, the corresponding gaussian random feature is defined as $[\sin(2\pi\omega t), \cos(2\pi\omega t)]$, where $[\cdot, \cdot]$ denotes the operation of feature concatenation. Such gaussian random feature can be used as an encoding for time step $t$ so that the score network can condition on $t$ by incorporating this encoding.

Table 1: Hyperparameter settings in IMDer.

| Hyperparameter | CMU-MOSI | CMU-MOSEI |
|---|---|---|
| Optimizer | Adam | Adam |
| Batch size | 32 | 128 |
| Learning rate | 0.001 | 0.002 |
| $\sigma$ used in our stochastic differential equation | 25 | 25 |
| Number of iterations for Euler–Maruyama solver | 500 | 500 |
| Shallow Feature Extractor | | |
| Kernel size for $\mathcal{E}_k$ | 3 | 3 |
| Hidden dimension for $\mathcal{E}_k$ | 32 | 32 |
| Missing Modality Diffused Network | | |
| Hidden dimensions for $s_k$ | $\{32, 64, 128, 256\}$ | $\{32, 64, 128, 256\}$ |
| Layers of cross-modal attention for $s_k$ | 2 | 2 |
| Number of attention heads for cross-modal attention | 8 | 8 |
| Hidden dimension for $\mathcal{D}_k$ | 64 | 64 |
| Number of RCAB for $\mathcal{D}_k$ | 20 | 20 |
| Multimodal Fusion and Prediction | | |
| Hidden dimension for $\mathcal{T}_k$ | 32 | 32 |
| Number of attention heads for $\mathcal{T}_k$ | 8 | 8 |
| Layers of transformer for $\mathcal{T}_k$ | 4 | 6 |

## 3 IMDer Settings

Tab. 1 lists the network architecture and the hyper-parameter settings in IMDer. We implemented all the experiments using PyTorch on an RTX 3090 GPU with 24GB memory. We explain the involved neural network components in IMDer as follows.

IMDer mainly consists of three parts: **shallow feature extractor**, **missing modality diffused network**, and **multimodal fusion and prediction**. First, we exploit a shallow feature extractor to encode the multimodal features $\mathbf{x}_k$ via separate 1D temporal convolutions $\mathcal{E}_k$, where $k \in \{l, v, a\}$. Second, we build a missing modality diffused network to model the modality-specific distribution of each missing modality. Typically, the modality distribution is captured via the score-based diffusion model $s_k$. As to achieve this, we sample from the modeled distribution space to recover the missing data. Then, we use the reconstruction module $\mathcal{D}_k$ to recover the final missing modalities. Each reconstruction module is composed of several residual channel attention blocks [7], where the 2D convolutional layers are replaced with 1D temporal convolutional layers to better fit the sequence

Table 2: Comparison of the unconditional and conditional recovering under random missing protocol. The values reported in each cell denote $ACC_2$/F1/$ACC_7$. MR: Missing Rate. **Bold** is the best.

| Datasets | MR | Unconditional Recovering | Conditional Recovering |
|---|---|---|---|
| | 0.0 | **85.7 / 85.6 / 45.3** | **85.7 / 85.6 / 45.3** |
| | 0.1 | 83.5 / 83.4 / 43.6 | **84.9 / 84.8 / 44.8** |
| | 0.2 | 81.6 / 81.3 / 41.6 | **83.5 / 83.4 / 44.3** |
| | 0.3 | 79.0 / 78.4 / 40.4 | **81.2 / 81.0 / 42.5** |
| CMU-MOSI | 0.4 | 76.4 / 75.8 / 38.1 | **78.6 / 78.5 / 39.7** |
| | 0.5 | 74.7 / 74.8 / 35.3 | **76.2 / 75.9 / 37.9** |
| | 0.6 | 72.3 / 72.3 / 33.2 | **74.7 / 74.0 / 35.8** |
| | 0.7 | 69.2 / 69.3 / 32.4 | **71.9 / 71.2 / 33.4** |
| | Average | 77.8 / 77.6 / 38.7 | **79.6 / 79.3 / 40.5** |
| | 0.0 | **85.1 / 85.1 / 53.4** | **85.1 / 85.1 / 53.4** |
| | 0.1 | 83.2 / 82.9 / 52.0 | **84.8 / 84.6 / 53.1** |
| | 0.2 | 81.3 / 80.4 / 50.7 | **82.7 / 82.4 / 52.0** |
| | 0.3 | 79.2 / 78.3 / 50.2 | **81.3 / 80.7 / 51.3** |
| CMU-MOSEI | 0.4 | 78.2 / 76.8 / 49.1 | **79.3 / 78.1 / 50.0** |
| | 0.5 | 77.6 / 75.7 / 47.8 | **79.0 / 77.4 / 49.2** |
| | 0.6 | 76.1 / 74.6 / 47.1 | **78.0 / 75.5 / 48.5** |
| | 0.7 | 75.4 / 73.6 / 46.0 | **77.3 / 74.6 / 47.6** |
| | Average | 79.5 / 78.4 / 49.5 | **80.9 / 79.8 / 50.6** |

features. Finally, the recovered modalities and the available modalities will be jointly fed into the multimodal transformers $\mathcal{T}_k$ [5] for feature fusion and MER.

## 4  Quantitative comparison of unconditional/conditional recovering

Besides the visualization results in the main file, we provide quantitative results of the unconditional and conditional recovering under the random missing protocol. Tab. 2 reports the quantitative results. Compared with the unconditional modality recovering mechanism, it is obvious that the conditional paradigm gains consistent performance improvements. This is reasonable as the available modalities will provide meaningful and semantic information to guide the modality recovering process, thus reducing the semantic ambiguity for the recovered modalities.

Table 3: MER accuracy comparison on CH-SIMS dataset under random missing protocol. The values reported in each cell denote $ACC_2$/F1/$ACC_5$. MR: Missing Rate. **Bold** is the best.

| Datasets | MR | MMIN | GCNet | **IMDer (Ours)** |
|---|---|---|---|---|
| | 0.0 | 74.9/75.0/45.3 | **76.7/76.8**/50.1 | 76.3/76.4/**50.7** |
| | 0.1 | 73.4/73.2/44.7 | 75.2/75.3/48.0 | **75.4/75.5/50.1** |
| | 0.2 | 71.9/71.5/44.1 | 73.7/73.6/45.1 | **74.7/74.5/47.3** |
| | 0.3 | 70.5/70.0/40.3 | 72.3/72.5/43.2 | **74.2/73.6/45.6** |
| CH-SIMS | 0.4 | 69.6/68.9/39.0 | 71.1/71.2/41.0 | **73.7/73.3/44.6** |
| | 0.5 | 67.1/67.1/37.0 | 69.7/69.4/40.5 | **72.7/72.3/43.3** |
| | 0.6 | 66.7/65.6/37.2 | 68.9/68.9/39.8 | **71.3/69.8/42.4** |
| | 0.7 | 64.3/63.8/35.1 | 67.8/67.8/35.9 | **69.8/69.6/42.2** |
| | Average | 69.8/69.4/40.3 | 71.9/71.9/43.0 | **73.5/73.1/45.8** |

## 5  Comparison on CH-SIMS

In this section, we consider a Chinese MER dataset CH-SIMS [6] to conduct further experiments. CH-SIMS contains 2281 refined video segments with fine-grained annotations of modalities. The data is collected from movies, TV serials, and variety shows. The annotation for each sample ranges from -1 (strongly negative) to 1 (strongly positive). For language modality, we extract the language features via pre-trained Chinese BERT model [2]. For vision modality, we use MultiComp OpenFace2.0 toolkit [1] to extract the set of 68 facial landmarks, 17 facial action units, head pose, head orientation,

and eye gaze. For audio modality, we use LibROSA speech toolkit [3] with default parameters to extract acoustic features. The experimental results are listed in the Tab. 3. Obviously, our proposed IMDer consistently achieves better results than MMIN or GCNet under random missing protocol.