# OpenReview forum: "Incomplete Multimodality-Diffused Emotion Recognition"
_NeurIPS.cc/2023/Conference — NeurIPS 2023 poster_

### Official Review · Reviewer_EQy6 · 2023-06-18

**Soundness:** 3 good
**Presentation:** 2 fair
**Contribution:** 2 fair
**Rating:** 4
**Confidence:** 5

**Summary:**

The authors recover missing modalities in multimodal emotion recognition by diffusion approaches. The test on two emotion databases and report SOTA results.

**Strengths:**

The usage of diffusion approaches is "trendy" and the recovery of missing data in MER is undercatered for, so the paper is particularly timely from a topic point of view.

**Weaknesses:**

From the abstract, it appears unclear, which modalities are involved, and whether these should be reconstructed for improved recognition (which seems to make limited sense, if the modality is actually missing) or for the sake of sheer synthesis. Even when reading the paper it is hard to capture what the modalities are, quickly, and ultimately, it seems sheer audiovisual (not including biosignals or symbolic information such as text), so the authors should rename to audiovisual. Also, reconstructions should have been visualised in the paper. A proper quantitative evaluation of results is absent, such as by effect sizes or proper significance testing including Bonferroni adjustment. The authors report Acc instead of the field's typical and more appropriate measure UAR. They fail to cite substantial key literature. The choice of data appears (too) homogeneous - better would be to use more different databases.

**Questions:**

Why do you think that this could generalise in real world applications rather than in quite self-similar data settings?

**Limitations:**

Affective Computing is at present heavily debated in terms of ELSI factors to the point where the EU considers its massive limitation in practice. This could be reflected by some thoughts.

---

> ### Author Rebuttal · Authors · 2023-08-09
>
> ## Thanks for recognizing our innovation and promising results, but you might have some misunderstanding. Below, we respond to each Question (Q) with an Answer (A).
>
> **Q1: From the abstract, it appears unclear, which modalities are involved, and whether these should be reconstructed for improved recognition (which seems to make limited sense, if the modality is actually missing) or for the sake of sheer synthesis. Even when reading the paper it is hard to capture what the modalities are, quickly, and ultimately, it seems sheer audiovisual (not including biosignals or symbolic information such as text), so the authors should rename to audiovisual.**
>
> A1: We appreciate your recognizing our motivation and contribution, but there should exist some misunderstandings. In the abstract, we had mentioned some modalities for MER in the first sentence, i.e., “Human multimodal emotion recognition...such as language, vision, and acoustic”. In particular, starting from Section 3.2 (e.g., Figure 2), we point out that we have considered the three heterogeneous modalities: language, vision, and acoustic. Therefore, it is not sheer audiovisual.
> For the question of whether these should be reconstructed for improved recognition, many prior works had indicated the effectiveness, more analysis is given in [a].
>
> [a] "GCNet: graph completion network for incomplete multimodal learning in conversation." IEEE TPAMI, 2023.
>
> **Q2: Also, reconstructions should have been visualised in the paper.**
>
> A2: In fact, Figure 3 and Figure 4 had provided the distribution visualization of reconstruction results. Please note the reconstruction is performed in a low-dimensional embedding space.
>
> **Q3: The choice of data appears (too) homogeneous - better would be to use more different databases. The authors report Acc instead of the field's typical and more appropriate measure UAR.**
>
> A3: Thanks for your suggestion. The MOSI and MOSEI datasets belong to the English dataset, but the MOSI focuses on movie reviews and MOSEI contains more than  250 topics. For your concern, we conduct extra experiments on another Chinese MER dataset CH-SIMS [b]. In addition, we add the UAR measure you mentioned for comparison. The experimental results are shown in the following table. Obviously, our proposed IMDer consistently achieves better results than the previous best methods: MMIN and GCNet under random missing protocol (MR means missing rate).
> |MR|MMIN|GCNet|Ours|
> |:----|:----|:----|:----|
> | |ACC$_2$/F1/ACC$_5$/UAR$_5$|ACC$_2$/F1/ACC$_5$/UAR$_5$|ACC$_2$/F1/ACC$_5$/UAR$_5$|
> |0.0|74.9/75.0/45.3/42.3|**76.7/76.8**/50.1/46.2|76.3/76.4/**50.7/**47.4****|
> |0.1|73.4/73.2/44.7/42.7|75.2/75.3/48.0/42.9|**75.4/75.5/50.1/46.1**|
> |0.2|71.9/71.5/44.1/41.1|73.7/73.6/45.1/42.5|**74.7/74.5/47.3/45.6**|
> |0.3|70.5/70.0/40.3/33.7|72.3/72.5/43.2/40.3|**74.2/73.6/45.6/45.6**|
> |0.4|69.6/68.9/39.0/32.1|71.1/71.2/41.0/40.5|**73.7/73.3/44.6/45.2**|
> |0.5|67.1/67.1/37.0/29.2|69.7/69.4/40.5/38.5|**72.7/72.3/43.3/45.3**|
> |0.6|66.7/65.6/37.2/28.9|68.9/68.9/39.8/36.5|**71.3/69.8/42.4/43.5**|
> |0.7|64.3/63.8/35.1/28.3|67.8/67.8/35.9/34.5|**69.8/69.6/42.2/38.9**|
> |Average|69.8/69.4/40.3/34.8|71.9/71.9/43.0/40.2|**73.5/73.1/45.8/44.7**|
>
>  [b] "Ch-sims: A chinese multimodal sentiment analysis dataset with fine-grained annotation of modality." ACL. 2020
>
> **Q4: A proper quantitative evaluation of results is absent, such as by effect sizes or proper significance testing including Bonferroni adjustment.**
>
> A4: Thanks for your suggestion. We consider the Cohen's d effect size to conduct new experiments on CH-SIMS dataset. The experimental results are shown in the following table, which proves that the performance improvement of our method is not minor. Especially in the 5-emotion class case, the performance improvement of our method is significant due to the fact that all effect sizes are greater than or equal to 0.8 according to prior rules [c].
> | Methods       |Cohen's d Effect sizes on ACC$_2$/F1/ACC$_5$/UAR$_5$  |
> |---------------|--------------------|
> | Ours vs. GCNet | 0.6 / 0.4 / 0.8 / 1.5|
> | Ours vs. MMIN  | 1.3 / 1.2 / 1.6 / 2.2|
>
> [c] Sawilowsky, Shlomo S. "New effect size rules of thumb." Journal of modern applied statistical methods 8.2 (2009): 26.
>
> **Q5: They fail to cite substantial key literature.**
>
> A5: Thanks for your suggestion.  We cannot understand what the substantial key literature refers to due to no any prompt. Of course, we may cite more literature related to multi-modal learning and emotion recognition, such as [d][e] in the revision.
>
> [d] "Multimodal machine learning: A survey and taxonomy." IEEE TPAMI, 2018.
>
> [e] "Emotion recognition from multiple modalities: Fundamentals and methodologies." IEEE Signal Processing Magazine, 2021.
>
> **Q6: Why do you think that this could generalise in real world applications rather than in quite self-similar data settings? Affective Computing is at present heavily debated in terms of ELSI factors to the point where the EU considers its massive limitation in practice. This could be reflected by some thoughts.**
>
> A6: Our method may be generalized into more real-world applications. First, all datasets we used come from real-world and not synthetic data, and the settings are standard. Second, the incomplete MER problem has been extensively studied in recent years, and recovering missing data could provide some extra cues for performance gains in real applications, more analysis is given in [a]. Third, to reduce the difficulty of recovery, we conduct modality recovery in low-dimensional feature distribution space, and pay more attention to the semantic consistency while recovering the modality.
> For your concern about the topic of Affective Computing, we think it is a rapidly developing open topic and increasing attention has been paid by researchers especially in recent years, please see those works in the Journal of IEEE Trans on Affective Computing. And, we believe the limitations could be gradually overcome in future research.

---

### Official Review · Reviewer_BscJ · 2023-07-05

**Soundness:** 3 good
**Presentation:** 3 good
**Contribution:** 3 good
**Rating:** 6
**Confidence:** 4

**Summary:**

This paper proposes a diffusion model-based generation method to solve the problem of missing modes in multimodal emotion recognition (MER), and in the model, the available modalities are utilized as prior conditions to guide and refine the recovering process to reduce the semantic ambiguity between the missing and the corresponding recovered modalities. Besides, quantitative experimental results verify that IMDer obtains state-of-the-art MER accuracy under various missing modality patterns.


**Strengths:**

# Originality
The authors use a score-based diffusion model to address the modal missingness in MER while using a cross-attention mechanism to guide and refine the recovery process by using other available modalities as a priori conditions to ensure that the recovered modalities are consistent in distribution and semantics.

# Quality
IMDer performs relatively well on each dataset, while the framework is neat and clear, making it a relatively effective and inspiring method.

# Clarity
The general idea and framework of this paper are relatively clearly described, but some of the simplifications in the formulas are not clearly described.

# Significance
Although the IMDer proposed in the article reaches SOTA, it is more significant to the community for two main reasons:
1. IMDer can not only be used to solve the modal deficiency problem but also be inspiring for other fields such as multimodal transformation and fusion.
2. It provides a mathematical derivation of the diffusion model for generating missing modalities.
Overall, the contribution of IMDer is significant.


**Weaknesses:**

Although this paper has achieved relatively good results, it cannot deal with the problem of intra-modal missing, e.g. some parts of the audio is missing. In addition, the ablation experiments are not sufficiently done, and other modules such as the semantic disambiguation module are not subject to ablation experiments. There is insufficient explanation of whether the simplification of the derivation of the formula is reasonable.

The experiments in the paper are insufficient, especially the ablation experiments, and the model comparison should be compared with the underlying methods cited in the Multimodal Fusion and Prediction module to demonstrate the effectiveness of the diffusion module.

The effectiveness of the semantic disambiguation part of diffusion is not evaluated, and the improvement of the recognition performance of Incomplete Multimodality cannot be assessed. It is not clear whether the improvement comes from the semantic information provided by other modalities during diffusion or rather from the extraction of better intra-modal information features.

Some closely related surveys and reviews on MER are missing, such as "Multimodal machine learning: A survey and taxonomy", and "Emotion Recognition From Multiple Modalities: Fundamentals and methodologies".


**Questions:**

Besides the above weakness, I also have the following questions:
Is it reasonable and valid for solving reverse-time SDE with some approximate scores? Does it limit the effect of the model?

**Limitations:**

No.

---

> ### Author Rebuttal · Authors · 2023-08-09
>
> ## Thanks for recognizing our innovation and promising results. Below, we respond to each Question (Q) with an Answer (A).
>
> **Q1: Although this paper has achieved relatively good results, it cannot deal with the problem of intra-modal missing, e.g. some parts of the audio is missing.**
>
> A1: The problem of intra-modal missing is a sequence or feature recovery task, and the objective is different from modal missing that we focus on. We are also interested in the problem of intra-modal missing and consider combining it with the modal missing problem by multi-task learning.
>
> **Q2: In addition, the ablation experiments are not sufficiently done, and  other modules such as the semantic disambiguation module are not subject to ablation experiments.**
>
> A2:  The ablation studies on possible cases have actually been provided in Section 4.3 in the manuscript and the supplementary material, including the semantic disambiguation. Concretely, we show the visualization results of the ablation experiments for semantic disambiguation (i.e., the conditional recovering) in Figure 4, and report the quantitative results of the ablation experiments for semantic disambiguation in Section 4 in the supplementary materials.
>
> **Q3: There is insufficient explanation of whether the simplification of the derivation of the formula is reasonable.**
>
> A3: There are two simplifications in our Problem Formulation. The first (in Eq.7) is
> $$p\_{t}\left(\mathbf{x}\_{\text{m}}(t) \middle| \mathbf{x}\_{\mathcal{I}\_{\text{obs}}}(t),\mathbf{x}\_{\mathcal{I}\_{\text{obs}}}(0) \right) \approx p\_{t}\left( \mathbf{x}\_{\text{m}}(t) \middle| \mathbf{x}\_{\mathcal{I}\_{\text{obs}}}(t) \right).$$
> For small $t$,  $\mathbf{x}\_{\mathcal{I}\_{\text{obs}}}(t)$ is almost the same as $\mathbf{x}\_{\mathcal{I}\_{\text{obs}}}(0)$ so the approximation holds. For large $t$,  $\mathbf{x}\_{\mathcal{I}\_{\text{obs}}}(0)$ becomes further away from $\mathbf{x}\_{\text{m}}(t)$ in the Markov chain, and thus have smaller impact on $\mathbf{x}\_{\text{m}}(t)$. Moreover, the approximation error for large $t$ have less effect on the final sample, since it is used early in the sampling process.
> Another simplification (in Eq.9) is
> $$\nabla\_{\mathbf{x}\_{\text{m}}}{\log{\mathbb{E}\_{p_{t}(\mathbf{x}\_{\mathcal{I}\_{\text{obs}}}{(t)}|\mathbf{x}\_{\mathcal{I}\_{\text{obs}}}{(0)})}\left\lbrack p\_{t}\left( \mathbf{x}\_{\text{m}}(t) \middle| \mathbf{x}\_{\mathcal{I}\_{\text{obs}}}(t) \right) \right\rbrack}} \approx \nabla\_{\mathbf{x}\_{\text{m}}}{\log{p\_{t}\left( \mathbf{x}\_{\text{m}}(t) \middle| \mathbf{x}\_{\mathcal{I}\_{\text{obs}}}(t) \right)}}.$$
> This is because the $\mathbf{x}\_{\mathcal{I}\_{\text{obs}}}(t)$ is a random sample from $p\_{t}\left(\mathbf{x}\_{\mathcal{I}\_{\text{obs}}}(t) \middle| \mathbf{x}\_{\mathcal{I}\_{\text{obs}}}(0) \right)$ in practice, so that we can simply this expectation term. We will reclarify the simplification according to your suggestion.
>
> **Q4: The experiments in the paper are insufficient, especially the ablation experiments, and the model comparison should be compared with the underlying methods cited in the Multimodal Fusion and Prediction module to demonstrate the effectiveness of the diffusion module.**
>
> A4: In fact, we had compared with the underlying method cited in the Multimodal Fusion and Prediction module in the Ablation study. As shown in Table 3, the method w/o $s\_k$ and $\mathcal{D}\_k$ denotes this underlying method, i.e., no modal recovery has been performed. Compared to the method w/ $s\_k$ the  effectiveness of diffusion module is proven.
>
> **Q5: The effectiveness of the semantic disambiguation part of diffusion is not evaluated, and the improvement of the recognition performance of Incomplete Multimodality cannot be assessed.**
>
> A5: For the question, you might miss our supplementary material. We had reported the quantitative results of the ablation experiments for semantic disambiguation (i.e., the conditional recovering) in Section 4 in the supplementary materials, and the results demonstrate the improvement of the recognition performance of Incomplete Multimodality.
>
> **Q6: It is not clear whether the improvement comes from the semantic information provided by other modalities during diffusion or rather from the extraction of better intra-modal information features.**
>
> A6: The improvement comes from two points:1. modality recovery (please see the ablation study in the Table3); 2. the semantic information provided by available modalities during diffusion (please see Section 4 in supplementary material).
>
> **Q7: Some closely related surveys and reviews on MER are missing, such as "Multimodal machine learning: A survey and taxonomy", and "Emotion Recognition From Multiple Modalities: Fundamentals and methodologies".**
>
> A7: Thanks for your suggestion. we will cite some closely related surveys and reviews on MER such as [a][b] you mentioned in the revision.
>
> [a] Baltrušaitis, Tadas, Chaitanya Ahuja, and Louis-Philippe Morency. "Multimodal machine learning: A survey and taxonomy." IEEE TPAMI, 2018.
>
> [b] Zhao, Sicheng, et al. "Emotion recognition from multiple modalities: Fundamentals and methodologies." IEEE Signal Processing Magazine, 2021.
>
> **Q8: Besides the above weakness, I also have the following questions: Is it reasonable and valid for solving reverse-time SDE with some approximate scores? Does it limit the effect of the model?**
>
> A8: The reasonable and valid for solving reverse-time SDE with some approximate scores have been well-proven by Song’s work, such as [c]. The approximation might limit the effect of the model to some degree, the exact density estimation on diffusion models is an open issue and beyond the scope of this work.
>
> [c] Song, Yang. Learning to Generate Data by Estimating Gradients of the Data Distribution. Ph.D thesis, Stanford University, 2022.

---

> > ### Comment · Reviewer_BscJ · 2023-08-17
> > **Final rating**
> >
> > Thank the authors for providing the rebuttal. My major concerns are addressed. I would like to keep my original rating.

---

### Official Review · Reviewer_bs2t · 2023-07-06

**Soundness:** 4 excellent
**Presentation:** 4 excellent
**Contribution:** 4 excellent
**Rating:** 8
**Confidence:** 4

**Summary:**

Paper proposes IMDer for multimodal emotion recognition under missing modalities. Language, vision, and audio modalities are evaluated using the CMU-MOSI and CMU-MOSEI datasets. Quantitative and qualitative results are shown including comparisons to state of the art, visualizations, and ablation study.

**Strengths:**

A nice approach to MER under missing modalities is proposed. The proposed approach is well motivated and detailed in paper. Contributions and extensions to state of the art are well defined in paper.

Evaluation is well conducted on two public datasets, showing encouraging results compared to state of the art. Fixed missing and random missing protocols are well done. Ablation study shows utility of both score model and reconstruction module. Qualitative results (Figure 3) look good showing closer distribution to ground truth compared to other approaches.

**Weaknesses:**

It is not clear how AUs are used as the video modality. OpenFace does not detect 35 different AUs. Is the AU feature vector AU occurrence and intensity? If so, does the occurrence add anything of value to it? An intensity of 0 would indicate the AU is not active.

Minor: There are typos that need fixed (e.g., "languate" on line 270).

**Questions:**

For the video modality, what exactly are the 35 AUs?

**Limitations:**

Limitations are adequately addressed.

---

> ### Author Rebuttal · Authors · 2023-08-09
>
> ## Thanks for recognizing our innovation and promising results. Below, we respond to each Question (Q) with an Answer (A).
>
> **Q1: It is not clear how AUs are used as the video modality. OpenFace does not detect 35 different AUs. Is the AU feature vector AU occurrence and intensity? If so, does the occurrence add anything of value to it? An intensity of 0 would indicate the AU is not active.**
>
> A1:  We apologize for the misunderstanding caused by the literature citation error that comes from the previous literature [a][b]. The literature [a][b] had been also quoted in our manuscript. For the question, we make a careful retrieval about the source, and further make the clarification as follows:
>
> The Facet is from iMotions [c] which is a commercial software toolkit not from OpenFace. Concretely, the vision modality we used is referred to the prior work [d] and is extracted by the CMU Multi-camp team (https://github.com/CMU-MultiComp-Lab/CMU-MultimodalSDK). For a video frame, the Facet is used to indicate 35 AUs that record the facial muscle movement for representing per-frame basic and advanced emotions (please see page 12 of [d]). For example,  for one of the face frames of a video clip, we extract the predicted values of 35 AUs that are the float value and not projected by activation function (i.e., the AUs value is not 0 or 1) as the feature vector of the face. If the video clip has 50 frames, then the tensor size of this vision modality is $50\times 35$.
>
> [a] "Decoupled Multimodal Distilling for Emotion Recognition." CVPR, 2023.
>
> [b] "Attention is not enough: Mitigating the distribution discrepancy in asynchronous multimodal sequence fusion." ICCV, 2021.
>
> [c] iMotions 2017. Facial expression analysis. [https://imotions.com/products/imotions-lab/modules/fea-facial-expression-analysis/](https://imotions.com/products/imotions-lab/modules/fea-facial-expression-analysis/)
>
> [d] "Multimodal transformer for unaligned multimodal language sequences." ACL, 2019.
>
> **Q2: Minor: There are typos that need fixed (e.g., "languate" on line 270).**
>
> A2: We will fix some typos in the revision according to your valuable suggestion.
>
> **Q3: For the video modality, what exactly are the 35 AUs?**
>
> A3: We use Facet from iMotions [c] to extract 35 facial action units (AUs) that record the facial muscle movement for representing basic and advanced emotions. The detailed definition of facial AUs may be obtained by decoding the Facet software. Here we follow the prior work [d] and the study (https://github.com/CMU-MultiComp-Lab/CMU-MultimodalSDK) of the CMU Multi-camp team. Given one of the face frames of a video clip, we extract the predicted values of 35 AUs that are the float value and not projected by activation function (i.e., the AUs value is not 0 or 1) as the feature vector of the face (more details are provided in A1).

---

> > ### Comment · Reviewer_bs2t · 2023-08-11
> >
> > Thank you for the clarification. My rating remains a strong accept.

---

### Official Review · Reviewer_4zZs · 2023-07-07

**Soundness:** 3 good
**Presentation:** 3 good
**Contribution:** 3 good
**Rating:** 6
**Confidence:** 4

**Summary:**

The paper presented an Incomplete Multimodality-Diffused emotion recognition (IMDer) method to mitigate the challenge of MER under incomplete multimodalities. To recover the missing modalities, IMDer exploited the score-based diffusion model that mapped the input Gaussian noise into the desired distribution space of the missing modalities and recovered missing data abided by their original distributions.

**Strengths:**

1. The proposed approach IMDer is applied to recover missing modalities.
2. The algorithm explanation contains a clear and detailed theory analysis.
3. Sufficient experiments and results.


**Weaknesses:**

1. Motivation is not clear for the proposed solution.
2. Major contributions are not well summarized.
3. Sec. 3.1 is too long, consider separating theory analysis into other sections.


**Questions:**

1.  Sec. 3.1 contains too much theory analysis. It is suggested to simplify or remove part of them to shorten the section.
2. In Figure 2, there is no connection between D_t and D_v. That may take confusion in understanding. Moreover, from D_a to hat(X)_a, there is no definition, which also leads to confusion.
3. In section 4.1, the Dataset part includes too much information. It is suggested to separate them into multiple parts which contain independent information, e.g. Datasets, Evaluation metrics….
4. In Table 2, there is MR: Missing Rate, but the missing rate involves which modality?
5. Some errors,
1) Line 185, with an common SDE…


**Limitations:**

See weakness.

---

> ### Author Rebuttal · Authors · 2023-08-09
>
> ## Thanks for recognizing our innovation and promising results. Below, we respond to each Question (Q) with an Answer (A).
>
> **Q1: Motivation is not clear for the proposed solution.**
>
> A1: The performance of MER in controlled scenarios has achieved satisfactory results in recent years. However, in real-world scenarios, not all modalities are consistently available, e.g., language data may be missing due to speech recognition errors; video data may be inaccessible due to privacy  and security concern. Ultimately, these incomplete multimodal data significantly hinder the performance of MER. Our main motivation is to preserve the performance of MER under missing modality cases as much as possible by recovering distributional and semantically consistent modality.
>
> **Q2: Major contributions are not well summarized.**
>
> A2: Our major contribution is to recover distribution-consistent missing modalities by the score-based diffusion models. In particular, to reduce the semantic ambiguity between the missing and the corresponding recovered modalities, we use the vanilla available modalities as the semantic conditions to guide and refine the recovering process. We will re-clarify this in the revision.
>
> **Q3: Sec. 3.1 contains too much theory analysis. It is suggested to simplify or remove part of them to shorten the section.**
>
> A3: Thanks for the suggestion. We will reorganize the theory analysis part by shortening this part in the revision.
>
> **Q4: In Figure 2, there is no connection between D_t and D_v. That may take confusion in understanding. Moreover, from D_a to hat(X)_a, there is no definition, which also leads to confusion.**
>
> A4: In Figure 2, we take the example that the acoustic modality is missing, thus there should not be connection between the two available modalities D_l (not “D_t”, typo by the reviewer) and D_v. In the caption of Figure 2, we had defined D_a to hat(X)_a: “Further, $\overset{\sim}{\text{x}}\_{a}(0)$ is fed into the reconstruction module $\mathcal{D}\_{a}$ to obtain the final recovered modality $\hat{\text{x}}\_{a}$”. For your concern, we will add the definition from D_a to hat(X)_a in Figure 2 for clarity.
>
> **Q5: In section 4.1, the Dataset part includes too much information. It is suggested to separate them into multiple parts which contain independent information, e.g. Datasets, Evaluation metrics….**
>
> A5:  Thanks for your suggestion. In the next revision, we will separate Dataset part into Datasets and Evaluation metrics parts for clarity.
>
> **Q6: In Table 2, there is MR: Missing Rate, but the missing rate involves which modality?**
>
> A6: MR measures the overall missingness ratio of the dataset, where the missing patterns are randomized for each sample, please see the description in lines 243-251.
>
> **Q7: Some errors, Line 185, with an common SDE…**
>
> A7: Thanks for your suggestion. We will polish the full paper and correct typos and grammar errors, including the article “a” in “with an common”.

---

> > ### Comment · Reviewer_4zZs · 2023-08-17
> >
> > Thanks for the detailed answer.
> > The authors addressed all of my concerns. I insist on my final decision, 6: Weak Accept.

---

### Official Review · Reviewer_tA7f · 2023-07-07

**Soundness:** 2 fair
**Presentation:** 3 good
**Contribution:** 3 good
**Rating:** 5
**Confidence:** 5

**Summary:**

This paper proposed a diffusion-based modality recovery mechanism to generate the missing modalities and then fuse the available modalities and the generated missing modalities for emotion prediction. The experiment results on MOSI and MOSEI shows that the proposed method can achieve state-of-the-art performance. However, there are some errors when introducing the related works, where the MCTN and MMIN aim to learn the multimodal joint representation through the cross-modality prediction strategy for missing modalities problems, not use the predicted modality for emotion prediction.

**Strengths:**

The paper proposed an efficient missing modality prediction method for missing modalities problems and reach the state-of-the-art performance.

**Weaknesses:**

1. There are some conceptual errors in this paper, shown in Summary Block.
2. The MOSI and MOSEI dataset are very similar, the author should explore more different datasets to verify the effectiveness of the proposed methods.
3. In Figure 4, the meaning of the different columns is confusing.

**Questions:**

Directly predicting the missing modalities signal or the sequence features of the missing modalities is very difficult based on the downstream dataset which only has thousands of samples, so the question is how to solve it and how is the prediction performance of missing modalities (for example, MSE/RMSE Loss).

**Limitations:**

As Weaknesses.

---

> ### Author Rebuttal · Authors · 2023-08-09
>
> ## Thanks for recognizing our innovation and promising results. Below, we respond to each Question (Q) with an Answer (A).
>
> **Q1: There are some conceptual errors in this paper, shown in Summary Block.**
>
> A1: For your question about some statement in the related work, we carefully check this part, it might exist some unclear statement and should not be conceptual errors. According to your suggestion,  we clarify the descriptions w.r.t. MTCN and MMIN: the MCTN and MMIN aim to learn the joint multimodal representation via the cross-modality recovery strategy with cycle consistency loss for missing modalities issue.
>
> **Q2: The MOSI and MOSEI dataset are very similar, the author should explore more different datasets to verify the effectiveness of the proposed methods.**
>
> A2:  Thanks for your suggestion. MOSI and MOSEI datasets are wildly used for multimodal emotion recognition. MOSI focuses on movie reviews and MOSEI contains more than  250 topics. For your concern, we conduct extra experiments on another Chinese MER dataset CH-SIMS [a]. The experimental results are shown in the following table. Obviously, our proposed IMDer consistently achieves better results than the previous best methods: MMIN and GCNet under random missing protocol (MR means missing rate).
> |MR|MMIN|GCNet|Ours|
> |:----|:----|:----|:----|
> | |ACC$_2$/F1/ACC$_5$|ACC$_2$/F1/ACC$_5$|ACC$_2$/F1/ACC$_5$|
> |0.0|74.9/75.0/45.3|**76.7**/**76.8**/50.1|76.3/76.4/**50.7**|
> |0.1|73.4/73.2/44.7|75.2/75.3/48.0|**75.4/75.5/50.1**|
> |0.2|71.9/71.5/44.1|73.7/73.6/45.1|**74.7/74.5/47.3**|
> |0.3|70.5/70.0/40.3|72.3/72.5/43.2|**74.2/73.6/45.6**|
> |0.4|69.6/68.9/39.0|71.1/71.2/41.0|**73.7/73.3/44.6**|
> |0.5|67.1/67.1/37.0|69.7/69.4/40.5|**72.7/72.3/43.3**|
> |0.6|66.7/65.6/37.2|68.9/68.9/39.8|**71.3/69.8/42.4**|
> |0.7|64.3/63.8/35.1|67.8/67.8/35.9|**69.8/69.6/42.2**|
> |Average|69.8/69.4/40.3|71.9/71.9/43.0|**73.5/73.1/45.8**|
>
> [a]  "Ch-sims: A chinese multimodal sentiment analysis dataset with fine-grained annotation of modality." ACL. 2020.
>
> **Q3: In Figure 4, the meaning of the different columns is confusing.**
>
> A3:  In Figure 4, each column illustrates a visualization comparison between unconditional (top row) and conditional (bottom row) recovery mechanism when a certain modality is missing. The missing modality for the first/second/third column is language/vision/acoustic modality, respectively. From Figure 4, we can conclude that conditional modality recovery mechanism shows the promising emotion category separability, indicating that using the available modalities as conditions can effectively reduce the semantic ambiguity. We will re-clarify the explanation in the revision.
>
> **Q4: Directly predicting the missing modalities signal or the sequence features of the missing modalities is very difficult based on the downstream dataset which only has thousands of samples, so the question is how to solve it and how is the prediction performance of missing modalities (for example, MSE/RMSE Loss).**
>
> A4: To reduce the difficulty and ambiguity of modality recovery, we conduct modality reconstruction in the low-dimensional feature distribution space. In addition, we pay more attention to the semantic consistency while recovering the modality, rather than just minimizing the MSE.  As to measure the prediction performance of missing modalities, in addition to the objective loss (i.e., score matching objective and MSE Loss), we have adopted 1) the accuracy of the downstream MER (Please see Table 1 and 2); 2) distribution visualization results for the recovery modality (Please see Figure 3 and 4).

---

> > ### Comment · Area_Chair_dqck · 2023-08-15
> > **Results with the CH-SIMS**
> >
> > Dear Authors,
> >
> > thanks for providing the additional results with CH-SIMS. I have a few questions:
> >
> > 1. I was wondering if you made any analysis on the stability of your results. For example, running the experiments on different data partitions, or just performing different training runs with different initializations on the same data partitions, and then computing standard deviations or confidence intervals.
> > 2. Another question I have is whether you made some qualitative analysis on the results. Did you observe any patterns on the instances that are well classified vs. instances where the model is not accurate?
> > 3. I'm also curious to know whether you made any interesting observations when you evaluated your method on the MOSI and MOSEI datasets (in english) vs. CH-SIMS (in Chinese).
> >
> > Thanks,
> >
> > AC

---

> > > ### Author Response · Authors · 2023-08-18
> > >
> > > Dear respected AC,
> > >
> > > Thanks for reading our rebuttal and concerns carefully.  Below, we respond to each Question (Q) with an Answer (A).
> > >
> > > **Q1: I was wondering if you made any analysis on the stability of your results. For example, running the experiments on different data partitions, or just performing different training runs with different initializations on the same data partitions, and then computing standard deviations or confidence intervals.**
> > >
> > > A1: Thanks for your suggestion. According to your suggestion, we perform *five training runs with different initializations on the same data partitions.* The standard deviations are reported in the following table. We observe that the standard deviations are in an acceptable range, which demonstrates the stability of our method.
> > >
> > > | Missing Rate | Standard Deviation  |
> > > |--------------|---------------------|
> > > |              | ACC$_2$/F1/ACC$_5$        |
> > > | 0.0          | 0.12/0.17/0.17      |
> > > | 0.1          | 0.08/0.21/0.45      |
> > > | 0.2          | 0.28/0.29/0.21      |
> > > | 0.3          | 0.29/0.25/0.22      |
> > > | 0.4          | 0.21/0.22/0.33      |
> > > | 0.5          | 0.17/0.29/0.21      |
> > > | 0.6          | 0.14/0.19/0.17      |
> > > | 0.7          | 0.17/0.33/0.22      |
> > >
> > >
> > >
> > >
> > > **Q2:  Another question I have is whether you made some qualitative analysis on the results. Did you observe any patterns on the instances that are well classified vs. instances where the model is not accurate?**
> > >
> > > A2: Thanks for your suggestion. We check the instance of this Chinese dataset, and observe that:
> > >
> > > i) The instances with consistent emotion in available modalities or clear emotion in some modalities, are often classified accurately. For example, a positive emotion sample is that the Chinese text description is “我很喜欢看你演的电影 (English meaning: I love watching your movies)”, and the facial expression and voice are very happy. When taking all modalities, or randomly dropping some modalities (but at least remaining one modality), our model can have an accurate prediction for these cases.
> > >
> > > ii) The instances with some hidden expressions (such as metaphor) might be misclassified. For example, a Chinese text description is “张扬，谢谢你 (English meaning: Zhang Yang, thank you)”, but we observe that the facial expression and voice are very angry. When all modalities are known, this instance can be accurately classified as a negative emotion label. But if only taking language, the model predicts as positive emotion; if taking language-visual, or language-acoustic, the model tends to misclassify them due to the hidden expression. It should be the limitation of our method as well as the current methods, which is expected to be solved in future research.
> > >
> > > **Q3: I'm also curious to know whether you made any interesting observations when you evaluated your method on the MOSI and MOSEI datasets (in english) vs. CH-SIMS (in Chinese).**
> > >
> > > A3: Thanks for your suggestion. According to the experiment results, we observe that the performance on the Chinese datasets is totally worse than the English datasets. The potential reason is that the emotion expression and perception can vary across different cultures and languages, especially, Chinese emotional expressions tend to be more implicit while English emotional expressions are relatively straightforward. This difference could impact the performance of emotion recognition models. Thus, it should be important to adapt the models to different linguistic and cultural environments in future.
> > >
> > > Thanks for your attention.

---

### Decision · Program_Chairs · 2023-09-21

**Decision:**

Accept (poster)

**Comment:**

This paper presents the In-complete Multimodality-Diffused emotion recognition method (IMDer) for multimodal emotion recognition under incomplete modalities. The method uses Gaussian noise in the input space and a score-based diffusion model to generate the data from the missing modalities. The experiments are performed on the CMU-MOSI and CMU-MOSEI benchmarks.

The reviewers valued the proposed technical approach and the reported state-of-the-art results in CMU-MOSI and CMU-MOSEI datasets. In particular, Reviewer BscJ considers the proposed IMDer method to be significant, since it could be inspiring for other problems, such as multimodal transformation/fusion. Besides some details on the presentation aspect (e.g. typos, clarifications about figures and related work, missing references, or the need for summarizing the major contributions), the reviewers highlighted some weaknesses on the evaluation. Concretely, three of the reviewers consider the experiments are not sufficient. In particular, Reviewer tA7F considers the two datasets used in the evaluation (CMU-MOSI and CMU-MOSEI) are very similar, and suggests the need for evaluation in more datasets. Authors reported additional experiments with the CH-SIMS dataset (a chinese multimodal sentiment analysis dataset) in their feedback. Their results on the CH-SIMS improve the best previous results reported on CH-SIMS.

This paper has been further discussed with the reviewers, AC, and SAC. The proposed method and formulation are interesting and solid. However, there are some important weaknesses that need to be addressed in the final version of the paper. Concretely,

1. More details on the datasets and the specific emotion recognition problem.

Authors should provide more details on the datasets used in their experiments (e.g. CMU-MOSEI and CMU-MOSI are collections of videos from youtube (talking heads) of people providing opinions, that have different human annotations, including valence (the emotional dimension that ranges from negative to positive) and emotion categories (these are available just for CMU-MOSEI)). Also, emotion recognition is a very broad topic. Authors should be more specific about the problem addressed in the paper (e.g. emotion signalling in videos from media).

2. Results on CH-SIMS

Authors should include the new results with CH-SIMS (which made some of the reviewers raise their scores and solved the concern on the homogeneity of the datasets used to evaluate the method). Also, authors should provide the technical details for the tests performed on CH-SIMS (for example, in the experiments on CMU-MOSEI and CMU-MOSI the authors used BERT to encode the text features, but authors did not specify how the text features are obtained for CH-SIMS).

3. More extended discussions on the limitations of the method.

The only limitations discussed in the paper are about the method, but nothing is said about the limitations of the method for emotion recognition use cases. For instance: (1) this method, trained with the datasets used in the paper, will probably not work to recognize emotion signalling in other scenarios (such as human-robot interaction), because emotion signalling might change depending on the context, culture, etc.; or (2) after training the method with CMU-MOSEI / CMU-MOSI, the resulting model will not perform well to perceive experienced emotion, since emotion experience and emotion signalling might differ.

4. Discussion on the ethical implications.

Since emotion recognition is a sensitive topic (as mentioned by Reviewer EQy6), a thoughtful discussion on the ethical implications of this work needs to be included in the paper. For guidelines on how to write the ethical implications discussion authors can check this document:

https://acii-conf.net/2023/wp-content/uploads/2023/03/instructions-ethical-statement.pdf

In particular, please check the following two sections: “Issues related to potential negative societal impact“ and “Issues related to limits of generalizability”.